# The capabilities of the adjoint of GEOS-Chem model to support HEMCO emission inventories and MERRA-2 meteorological data

Zhaojun Tang[1], Zhe Jiang[1]*, Jiaqi Chen[1], Panpan Yang[1], Yanan Shen[1]

[1]School of Earth and Space Sciences, University of Science and Technology of China, Hefei, Anhui, 230026, China.

*Correspondence to: Zhe Jiang (zhejiang@ustc.edu.cn)

## Abstract

Adjoint of the GEOS-Chem model has been widely used to constrain the sources of atmospheric compositions. Here we designed a new framework to facilitate emission inventory updates in the adjoint of GEOS-Chem model. The major advantage of this new framework is good readability and extensibility, which allows us to support Harmonized Emissions Component (HEMCO) emission inventories conveniently and to easily add more emission inventories following future updates in GEOS-Chem forward simulations. Furthermore, we developed new modules to support MERRA-2 meteorological data, which allows us to perform long-term analysis with consistent meteorological data in 1979-present. The performances of the developed capabilities were evaluated with the following steps: 1) diagnostic outputs of carbon monoxide (CO) sources and sinks to ensure the correct reading and use of emission inventories; 2) forward simulations to compare the modeled surface and column CO concentrations among various model versions; 3) backward simulations to compare adjoint gradients of global CO concentrations to CO emissions with finite difference gradients; and 4) observing system simulation experiments (OSSE) to evaluate the model performance in 4D variational (4D-var) assimilations. Finally, an example application of 4D-var assimilation was presented to constrain anthropogenic CO emissions in 2015 by assimilating Measurement of Pollution in the Troposphere (MOPITT) CO observations. The capabilities developed in this work are important for better applications of the adjoint of GEOS-Chem model in the future.

These capabilities will be submitted to the standard GEOS-Chem adjoint code base for better
development of the community of the adjoint of GEOS-Chem model.
**1. Introduction**
GEOS-Chem is a global 3D chemical transport model (CTM) and has been widely used
to analyze the sources and variabilities of atmospheric compositions (Whaley et al., 2015; Li
et al., 2019; Hammer et al., 2020; Jiang et al., 2022). GEOS-Chem model is driven by
meteorological reanalysis data from the Goddard Earth Observing System (GEOS) of the
Global Modeling and Assimilation Office (GMAO). Emissions in GEOS-Chem model are
calculated with state-of-the-art inventories such as CEDS (Community Emissions Data
System) (Hoesly et al., 2018), MIX (Li et al., 2017) and NEI2011 (National Emissions
Inventory). Based on GEOS-Chem forward simulation, the adjoint of the GEOS-Chem model
(Henze et al., 2007) further provides the capability of backward simulation of physical and
chemical processes within the 4D variational (4D-var) framework. The major advantage of the
adjoint model is obtaining the sensitivity of atmospheric concentrations to multiple model
variables within a single backward simulation. The major applications of the adjoint of GEOS-
Chem model include inverse analyses of atmospheric composition emissions by minimizing
the difference between simulations and observations (Jiang et al., 2015a; Zhang et al., 2018;
Qu et al., 2022) as well as sensitivity analyses to analyze the sources of atmospheric
compositions (Jiang et al., 2015b; Zhao et al., 2019; Dedoussi et al., 2020).
The algorithm of the 4D-var framework requires identical model processes in the forward
and backward simulations. Ideally, the code for the adjoint model should be updated following
the GEOS-Chem forward codes to take advantage of the new features in GEOS-Chem forward
simulations. However, the updates in the adjoint model are difficult and usually delayed. For
example, the MEERA-2 meteorological reanalysis data with temporal coverage of 1979-
present were supported in the GEOS-Chem forward simulations in v11-01. The adjoint of
GEOS-Chem model does not support MERRA-2, and thus, long-term analysis must combine
different meteorological reanalysis data, such as GEOS-4 (1985-2007), GEOS-5 (2004-2012)
and GEOS-FP (2012-present). For instance, Jiang et al. (2017) constrained global carbon
monoxide (CO) emissions in 2001-2015, while the derived trends in CO emissions in Jiang et
al. (2017) could be affected by the discontinuity among various versions of the meteorological
data (i.e., GEOS-4 in 2001-2003, GEOS-5 in 2004-2012 and GEOS-FP in 2013-2015) and the
lack of consistency in the model physics of GEOS-5.

Emission inventories play a key role in the simulation of atmospheric compositions.

Harmonized Emissions Component (HEMCO) (Keller et al., 2014; Lin et al., 2021) was
included in the GEOS-Chem forward simulations in v10-01. HEMCO is responsible for inputs
of meteorological and emission data with default support for emission inventories such as
CEDS, MIX and NEI2011. New emission inventories can be added readily within HEMCO
framework. There are noticeable differences between HEMCO and the adjoint of GEOS-Chem
model. First, meteorological and emission data are read with individual modules in the adjoint
of GEOS-Chem model. Second, the inputs of emission inventories are undertaken by different
modules that were developed individually with significant discrepancies in the source code. In
addition, the file format (e.g., binary punch in the adjoint of GEOS-Chem that is the format of
older GEOS-Chem versions in contrast to netCDF in HEMCO), emission variables and the
usage methods of emission variables (e.g., emission hierarchy, scaling factors and time slice)
are inconsistent. These differences have posed a barrier to the application of new emission
inventories in the adjoint of GEOS-Chem model.

The lack of support to the updated emission inventories can affect the applications of the

adjoint of GEOS-Chem model. First, adjoint-based sensitivity analyses are obtained by the
backward simulations of atmospheric compositions (i.e., adjoint tracers) and the combination
of adjoint tracers with emissions. Out-of-date emission inventories can thus result in inaccurate
estimation of the adjoint sensitivities. Second, while inverse analyses are constrained by
atmospheric observations, the updated emission inventories are still critical because they are
helpful for better convergence of 4D-var assimilations by setting a more reasonable a priori
penalty in the cost function. For instance, the a priori biomass burning CO emissions (GFED3,
van der Werf et al. (2010)) in Jiang et al. (2017) lack interannual variabilities later than 2011.
In order to obtain reasonable convergence of biomass burning emissions, the a priori biomass
burning emissions in September-November 2006 were applied to September-November 2015
over Indonesia in Jiang et al. (2017).
Ideally, people should consider porting the complete HEMCO to the adjoint of GEOS-
Chem model to match the new features in GEOS-Chem forward simulations. However, a
complete port of HEMCO implies replacing the input framework of the adjoint of GEOS-Chem
model, as well as restructuring of HEMCO and the adjoint of GEOS-Chem model to address
the compatibility issues, which is very challenging and may not be necessary because the
meteorological modules still work well in the adjoint of GEOS-Chem model. Consequently, a
major objective of this work is to design a new framework to facilitate emission inventory
updates in the adjoint of GEOS-Chem model. For this objective, this new framework must have
good readability and extensibility to allow us to support HEMCO emission inventories
conveniently and to add more emissions inventories following future updates in GEOS-Chem
forward simulations easily. Furthermore, we developed new modules to support MERRA-2
meteorological data within the current framework of the adjoint of GEOS-Chem model, as
reuse of existing frameworks can save much work.
CO is one of the most important atmospheric pollutants and plays a key role in
tropospheric chemistry. Sources of atmospheric CO include fossil fuel combustion, biomass
burning and oxidation of hydrocarbons. The major sink of atmospheric CO is hydroxyl
radical  (OH). The simple chemical sink of atmospheric CO allows us to simulate atmospheric
CO with linearized chemistry; for example, the tagged-CO mode of the GEOS-Chem model
can reduce the calculation cost by 98% with respect to the full chemistry mode by reading
archived monthly OH fields. The tagged-CO mode of the GEOS-Chem model has been widely
used to investigate the sources and variabilities of atmospheric CO in recent decades (Heald et
al., 2004; Kopacz et al., 2009; Jiang et al., 2017). The capabilities developed in this work are
thus based on the tagged-CO mode, as it can effectively accelerate the model development
process. More efforts are needed in the future to extend these capabilities to support emissions
inventories associated with full chemistry simulations.
The results presented in this paper show the development, integration, evaluation, and
application of these new capabilities, which is important to better applications of the adjoint of
GEOS-Chem model in the future. The capabilities developed in this work will be submitted to
the standard GEOS-Chem adjoint code base (Henze et al., 2007) for better development of the
community of the adjoint of GEOS-Chem model. This paper is organized as follows: in Section
2, we describe the adjoint of GEOS-Chem model, the development of these new capabilities,
and the Measurement of Pollution in the Troposphere (MOPITT) CO observations used in this
work. In Section 3, we evaluated the performances of the developed capabilities in forward and
backward simulations, together with observing system simulation experiments (OSSE) to
evaluate the model performance in 4D-var assimilations. An example application of 4D-var
assimilation to constrain anthropogenic CO emissions in 2015 by assimilating MOPITT CO
observations was also presented. Our conclusions follow in Section 4.

**2. Methodology and Data**
**2.1 Adjoint of the GEOS-Chem model**
We use version v35n of the adjoint of GEOS-Chem model. Our analysis is conducted at
a horizontal resolution of 4°×5° with 47 vertical levels and employs the CO-only simulation
(tagged-CO mode). The global default anthropogenic emission inventory in the standard
version of the adjoint of GEOS-Chem model (hereafter referred to as GC-Adjoint-STD) is
Global Emissions InitiAtive (GEIA), but is replaced by the following regional emission
inventories: NEI2008 in North America, the Criteria Air Contaminants (CAC) inventory for
Canada, the Big Bend Regional Aerosol and Visibility Observational (BRAVO) Study
Emissions Inventory for Mexico (Kuhns et al., 2003), the Cooperative Program for Monitoring
and Evaluation of the Long-range Transmission of Air Pollutants in Europe (EMEP) inventory
for Europe in 2000 (Vestreng and Klein, 2002) and the INTEX-B Asia emissions inventory for
2006 (Zhang et al., 2009). Biomass burning emissions are based on the GFED3 (van der Werf
et al., 2010).

The objective of the 4D-var approach is to minimize the difference between simulations

and observations described by the cost function (Henze et al., 2007):
$$J(\boldsymbol{x}) = \sum_{i=1}^{N}(\boldsymbol{F}_i(\boldsymbol{x}) - \boldsymbol{z}_i)^T \boldsymbol{S}_{\Sigma}^{-1}(\boldsymbol{F}_i(\boldsymbol{x}) - \boldsymbol{z}_i) + \gamma(\boldsymbol{x} - \boldsymbol{x}_a)^T \boldsymbol{S}_a^{-1}(\boldsymbol{x} - \boldsymbol{x}_a)$$ (1)
where $\boldsymbol{x}$ is the state vector of CO emissions, $N$ is the number of observations that are
distributed in time over the assimilation period, $\boldsymbol{z}_i$ is a given measurement, and $\boldsymbol{F}(\boldsymbol{x})$ is the
forward model. The error estimates are assumed to be Gaussian and are given by $\boldsymbol{S}_{\Sigma}$, the
observational error covariance matrix, and $\boldsymbol{S}_a$, the a priori error covariance matrix. The cost
function is minimized through minimizing the adjoint gradients by adjusting the CO emissions
iteratively:
$$\nabla_x J(\boldsymbol{x}) = \sum_{k=1}^{N}\left[2\boldsymbol{S}_{\Sigma}^{-1}(\boldsymbol{F}_i(\boldsymbol{x}) - \boldsymbol{z}_i)\frac{\partial F_i}{\partial x}\right] + 2\gamma \boldsymbol{S}_a^{-1}(\boldsymbol{x} - \boldsymbol{x}_a)$$ (2)
We assume a uniform observation error of 20%. The combustion CO sources (fossil fuel,
biofuel and biomass burning) and the oxidation source from biogenic volatile organic
compounds (VOCs) are combined, assuming a 50% uniform a priori error. We optimize the
source of CO from the oxidation of methane ($CH_4$) separately as an aggregated global source,
assuming an a priori uncertainty of 25%. The CO emission estimates are optimized with
monthly temporal resolution. Following Jiang et al. (2017), we performed 40 iterations
(forward + backward simulations) for each month, which usually produced 6-8 accepted
iterations (i.e., successful line searches in the large-scale bound constrained optimization (L-
BFGS-B, Zhu et al. (1997)) to reduce the cost functions and adjoint gradients. The a posteriori
CO emission estimates were calculated based on the last accepted iteration, which usually
corresponded to the iteration with the lowest cost function.
**2.2 New framework to read emission inventories**
A major objective of this work is to design a new framework to facilitate emission
inventory updates in the adjoint of GEOS-Chem model. As shown in Fig. 1, we first initialize
the array in [INITIAL] and batch read the emission data in [READ_DATA], which were
interpolated offline with $1°×1°$ resolution by considering the mass conservation. Here, the data
include the emission inventory data listed in Table S1 (see the SI), the corresponding scaling
factor data and the mask map files of domain definitions. The data are scaled in
[SCALE_DATA] by multiplying the corresponding annual, season, month, week, and 24-hour
emission factors and are then online interpolated to the current resolution ($4°×5°$ in this work)
of the model by [RGRID_DATA], which was followed by the application of region masks in
[MASK].
The emission variable of CO obtained in this part is written to the model memory in
emission.f and emission_adj.f by calling DO_EMISSIONS to ensure the consistent emissions
in both forward and backward simulations. The GET_[TRACER] subroutines are used to
obtain the CO emission variable, which participates in the calculation of physicochemical
processes in the model, to interact with other modules. Finally, the variable is cleaned from the
memory by the [CLEANUP] module. It should be noted that a two-step interpolation is
employed in this work (hereafter referred to as GC-Adjoint-HEMCO) following GC-Adjoint-
STD, for example, 0.1°×0.1° to 1°×1° and then to 4°×5° for the NEI2011 inventory, which is
different from the one-step interpolation in GEOS-Chem forward model (v12-08-01, hereafter
referred to as GC-v12), for example, 0.1°×0.1° to 4°×5° directly for the NEI2011 inventory.
The different interpolation methods can lead to differences in the interpolated emission data.
**2.3 Updates in emission inventories**
In addition to baseline emission data, there are critical factors that affect the usage of
emission data in the models. Reading the emission data correctly thus does not necessarily
mean using emission data correctly. For example, emission hierarchy is used to prioritize
emission fields within the same emission category. Emissions of higher hierarchy overwrite
lower hierarchy data. Regional emission inventories usually have a higher hierarchy within
their mask boundaries. Scaling factors are used to adjust the baseline emissions with annual,
season, month, week, and 24-hour temporal scales. Time slice selection is used to define the
usage methods of the emission data outside the original temporal range; for instance, data can
be interpreted as climatology and recycled once the end of the last time slice is reached or be
only considered as long as the simulation time is within the time range. Consequently, we must
validate the integrated emissions carefully to ensure that the abovementioned factors have been
correctly applied and to ensure that the calculated emissions are reasonable for individual
inventories and the combination of all inventories.
To take advantage of this new framework, six HEMCO emission inventories have been
added to this work. To validate the emissions, we performed actual simulations with GC-v12,
GC-Adjoint-HEMCO and GC-Adjoint-STD, and the emissions were calculated in the model
simulations and then output to the Log file. As shown in Table S1, the CEDS emission
inventory (0.5°×0.5°) is adopted in GC-Adjoint-HEMCO to provide global default emissions
for 1750-2019. The diurnal scale factors are applied to obtain CO emissions at different
moments of the day. Fig. S1 (see the SI) shows CEDS CO emissions in 2015 in GC-v12 and
GC-Adjoint-HEMCO and GEIA CO emissions in GC-Adjoint-STD, and we find noticeable
differences in CO emissions between CEDS and GEIA. As shown in Table 1, the CEDS CO
emissions in 2015 were 613.57 and 613.85 Tg/y in GC-v12 and GC-Adjoint-HEMCO,
respectively, with a relative difference of 0.05% between GC-v12 and GC-Adjoint-HEMCO.
The GEIA CO emissions in 2015 were 445.88 Tg/year in GC-Adjoint-STD.
The default CEDS inventory is replaced by the following regional emission inventories
in GC-Adjoint-HEMCO: MIX in Asia (0.25°×0.25°), NEI2011 in the United States
(0.1°×0.1°), DICE_AFRICA and EDGARV43 in Africa (0.1°×0.1°) and APEI in Canada
(0.1°×0.1°). As shown in Fig. S2 (see the SI), the MIX inventory provides Asian emissions in
2008-2010, accompanied by diurnal scale factors to describe daily emission variation. The
1°×1° scale factors in the AnuualScalar.geos.1x1.nc file further provide the annual variation in
1985-2010. As shown in Table 1, the MIX CO emissions in 2015 were 321.18 and 321.71 Tg/y
in GC-v12 and GC-Adjoint-HEMCO, respectively, with a relative difference of 0.17% between
GC-v12 and GC-Adjoint-HEMCO. The INTEX-B CO emissions in 2015 were 353.03 Tg/y in
GC-Adjoint-STD.
The NEI2011 inventory (Fig. S3, see the SI) provides anthropogenic emissions for the
United States in 2011 with annual scalar factors from 2006-2013. The weekday and weekend
factors are read from NEI99.dow.geos.1x1.nc file since 1999 with all CO factors of 1.0 on
weekdays and between 0.990 and 0.997 on Saturdays and Sundays. The NEI2011 CO
emissions in 2015 were 35.83 and 37.70 Tg/y in GC-v12 and GC-Adjoint-HEMCO,
respectively, with a relative difference of 5.22% between GC-v12 and GC-Adjoint-HEMCO.
The NEI2008 CO emissions in 2015 were 52.87 Tg/y in GC-Adjoint-STD. APEI (Fig. S4, see
the SI) is the primary source of anthropogenic emissions in the Canadian domain. The APEI
CO emissions in 2015 were 6.10 and 6.17 Tg/y in GC-v12 and GC-Adjoint-HEMCO,
respectively, with a relative difference of 1.14% between GC-v12 and GC-Adjoint-HEMCO.
The CAC CO emissions in 2015 were 10.20 Tg/y in GC-Adjoint-STD. Following GC-v12, the
CO emissions in APEI are enhanced by 19% to account for coemitted VOC in the tagged-CO
simulation.

Emissions for the African domain are provided by the combination of DICE_AFRICA

and EDGARV43 (Fig. S5, see the SI). Here DICE_AFRICA includes anthropogenic and
biofuel emissions in 2013. We read the DICE_AFRICA emissions data into the model in two
types according to the guidelines of the inventory. Emissions from sectors such as automobiles
and motorcycles are aggregated into anthropogenic sources, and household-generated
emissions such as charcoal and agricultural waste are aggregated into biofuel sources. Efficient
combustion emissions from EDGAR v4.3 in 1970-2010 then compensate for the lacking
sources in DICE_AFRICA. Daily variation factors for CO are also used here for emissions
across the African region. The 2010 CO seasonal scale factors are used in EDGAR v4.3 for
sectoral emission sources. The DICE_AFRICA and EDGARV43 CO emissions in 2015 were
83.42 and 83.02 Tg/y in GC-v12 and GC-Adjoint-HEMCO, respectively, with a relative
difference of -0.48% between GC-v12 and GC-Adjoint-HEMCO. Following GC-v12, the CO
emissions in DICE_AFRICA and EDGARV43 are enhanced by 19% to account for coemitted
VOC in the tagged-CO simulation.

The biomass burning emission inventory in GC-Adjoint-HEMCO is GFED4 (Fig. S6,

see the SI), which includes dry matter emissions from a total of seven sectors in 1997-2019.
The same GFED_emssion_factors.H header file as in the GC-v12 version is read in the GC-
Adjoint-HEMCO. This file contains the ratio factors of atmospheric pollutants, and we
multiply the ratio factors one by one according to the ID of each species to ensure that the
species in the model have biomass burning sources. The GFED4 CO emissions in 2015 were
437.13 and 435.89 Tg/y in GC-v12 and GC-Adjoint-HEMCO, respectively, with a relative
difference of -0.28% between GC-v12 and GC-Adjoint-HEMCO. The GFED3 CO emissions
in 2015 were 382.04 Tg/year in GC-Adjoint-STD. Following GC-v12, the combustion CO
sources in biomass burning are enhanced by 5% to consider the CO generated by VOC in the
tagged-CO simulation.

Fig. 2 shows the total combustion CO emissions in 2015 from GC-v12, GC-Adjoint-

HEMCO and GC-Adjoint-STD. As shown in Table 2, the regional combustion CO emissions
are 320.66 and 320.38 Tg/y (Asia), 73.96 and 66.93 Tg/y (North America), 199.51 and
193.29/y Tg (Africa), 79.04 and 78.91 Tg/y (South America), 31.58 and 30.96 Tg/y (Europe)
and 12.24 and 11.99 Tg/y (Australia) in GC-v12 and GC-Adjoint-HEMCO, respectively. Fig.
3 further shows the monthly combustion CO emissions in 2015 from GC-v12, GC-Adjoint-
HEMCO and GC-Adjoint-STD, and there are good agreements in the monthly variation of CO
emissions between GC-v12 and GC-Adjoint-HEMCO. The CO emissions in GC-Adjoint-STD
are similar to those in GC-v12 and GC-Adjoint-HEMCO in winter and spring but with large
differences in summer and autumn. This seasonal difference may reflect the influence of
different emission inventories on biomass burning.
**2.4 Updates in CO chemical sources and sinks**

The biogenic emissions in GC-Adjoint-STD are Model of Emissions of Gases and

Aerosols from Nature, version 2.0 (MEGANv2.0, Guenther et al. (2006)) in the full chemistry
simulation but are GEIA in the tagged-CO simulation (Fig. S7, see the SI). Fisher et al. (2017)
demonstrated improvement in modeled CO concentrations in tagged-CO simulation by reading
archived VOC- and $CH_4$-generated CO fields provided by full chemistry simulation. The
archived VOC- and $CH_4$-generated CO fields in 2013 (PCO_3Dglobal.geosfp.4x5.nc) were set
as the default CO chemical sources in the tagged-CO simulation in GC-v12 and supported in
GC-Adjoint-HEMCO. As shown in Table 2, the CO chemical sources (columns) obtained by
reading the archived VOC- and $CH_4$-generated CO fields demonstrate good agreement between
GC-v12 and GC-Adjoint-HEMCO. However, they are 30-60% lower than those in GEIA in
GC-Adjoint-STD, and this difference could be partially associated with the inconsistency
between the archived VOC-generated CO fields in 2013 and the actual meteorological data in
2015 in the simulation.

The default $CH_4$-generated CO emissions in GC-Adjoint-STD (Fig. S8, see the SI) are

calculated based on averaged $CH_4$ concentrations in four latitude bands (90°S - 30°S, 30°S -
00°S, 00°N - 30°N, 30°N - 90°N), which are based on Climate Monitoring and Diagnostics
Laboratory (CMDL) surface observations and Intergovernmental Panel on Climate Change
(IPCC) future scenarios. As shown in Table 2, there are good agreements in the $CH_4$-generated
CO emissions between GC-v12 and GC-Adjoint-HEMCO by reading
PCO_3Dglobal.geosfp.4x5.nc, and they are 20-60% lower than those in CMDL/IPCC in GC-
Adjoint-STD. Furthermore, the default archived monthly OH fields were updated following
GC-v12 with updated calculations for the decay rate (KRATE, from JPL 03 to JPL 2006) in
GC-Adjoint-HEMCO. The subsequent CO sinks (Fig. S9, see the SI) in GC-v12 and GC-
Adjoint-HEMCO are 20-40% higher than those in GC-Adjoint-STD.
**2.5 Updates in meteorological data**

The MERRA-2 meteorological data (1979-present) are supported in GC-Adjoint-

HEMCO to ensure long-term consistency in the meteorological data in the analyses. The code
porting to support MERRA-2 follows the current framework of the adjoint of GEOS-Chem
model, particularly because the meteorological variables and vertical resolutions of MERRA-
2 are the same as those of GEOS-FP (2012-present), while GEOS-FP is already supported by
GC-Adjoint-STD. Fig. 4A-B show the averages of surface CO concentrations in 2015 from
GC-Adjoint-HEMCO driven by MERRA-2 and GEOS-FP, respectively. Our results
demonstrate lower surface CO concentrations driven by MERRA-2 (Fig. 4C), although there
is good agreement in the spatial distributions of CO concentrations. Similarly, Fig. 4D-F show
the averages of CO columns in 2015 from GC-Adjoint-HEMCO driven by MERRA-2 and
GEOS-FP and their differences. Despite the noticeable differences in surface CO
concentrations (Fig. 4C), the differences in CO columns (Fig. 4F) are much smaller, and the
modeled CO columns driven by MERRA-2 are higher than those driven by GEOS-FP over the
Indian Ocean. The discrepancy between surface and column CO in Fig. 4 may reflect the
impacts of different convective transports on the modeled CO concentrations.
**2.6 MOPITT CO measurements**

The MOPITT data used here were obtained from the joint retrieval (V7J) of CO from

thermal infrared (TIR, 4.7μm) and near-infrared (NIR, 2.3μm) radiances using an optimal
estimation approach (Worden et al., 2010; Deeter et al., 2017). The retrieved volume mixing
ratios (VMR) are reported as layer averages of 10 pressure levels with a footprint of 22 km $\times$
22 km. Following Jiang et al. (2017), we reject MOPITT data with CO column amounts less
than $5\times10^{17}$ molec/cm$^2$ and with low cloud observations. Since the NIR channel measures
reflected solar radiation, only daytime data are considered.

# 3. Model evaluation and application
**3.1 Model performances in forward and backward simulations**

The reasonable emissions in the diagnostic outputs in Section 2 do not necessarily mean

the correct integration of emissions in the assimilations. Consequently, here we evaluate the
performance of GC-Adjoint-HEMCO in forward simulations. Fig. 5 shows the averages of
surface and column CO concentrations in 2015 from GC-v12, GC-Adjoint-HEMCO and GC-
Adjoint-STD. As shown in Table 2, the regional differences between GC-v12 and GC-Adjoint-
HEMCO are 2.6%, -5.7%, -4.6%, -1.7%, -1.4% and -3.6% in surface CO concentrations, and
-2.3%, -3.6%, -3.3%, -3.1%, -3.3% and -4.1% in CO columns over Asia, North America,
Africa, South America, Europe, and Australia, respectively. There are larger regional
differences in CO concentrations between GC-v12 and GC-Adjoint-STD: 4.6%, -10.1%, 6.3%,
22.5%, 6.4% and 25.7% in surface CO concentrations, and -0.7%, -9.9%, 2.5%, 8.0%, -5.8%
and 8.5% in CO columns over Asia, North America, Africa, South America, Europe, and
Australia, respectively. The agreement between GC-v12 and GC-Adjoint-HEMCO confirms
the reliability of GC-Adjoint-HEMCO in forward simulations, while the small differences in
CO concentrations between GC-v12 and GC-Adjoint-HEMCO are expected in view of the
comparable differences in regional emissions, chemical sources and sinks, as shown in Table

2.

In addition to forward simulations, the reliability of 4D-var assimilation also relies on

the accuracy of the adjoint-based sensitivities, which are obtained by the backward simulations
of adjoint tracers and the combination of adjoint tracers with emissions. As mentioned in
Section 2.2, we have made corresponding modifications to both forward and backward
modules. Consequently, here we further evaluate the performance of GC-Adjoint-HEMCO in
backward simulations. Here the adjoint gradients are simplified as:
$$\nabla_x J(x) = \frac{\partial F_N}{\partial x} \tag{3}$$

The adjoint gradients (Eq. 3) represent the sensitivities of modeled atmospheric compositions
at the final time step (i.e., $i = N$) to emissions, which were then compared with the finite
difference gradients calculated with:
$$\Lambda = \frac{J(x+\delta x) - J(x-\delta x)}{2\delta x} \tag{4}$$

Here the finite difference gradients represent the response of modeled atmospheric
compositions at the final time step to finite perturbations in emissions provided by the forward
simulations ($\delta x = 10\%$ in this work).

Fig. 6A-C show the comparison of adjoint and finite difference gradients of global

surface CO concentrations to CO emissions with a 24-hour assimilation window by turning on
the convection, planetary boundary layer mixing and advection processes individually. We find
good consistency in the gradients with respect to convection and planetary boundary layer
(PBL) mixing. The larger deviation with respect to advection is caused by the discrete
advection algorithm in forward simulations and continuous advection algorithm in backward
simulations (Henze et al., 2007). Fig. 6D-F further exhibit the effects of combined model
processes (turning off advection as suggested by Henze et al. (2007)). We find good agreement
between the adjoint and finite difference gradients with different assimilation windows (24
hours, 7 days and one month). Moreover, Fig. S10 and S11 (see the SI) demonstrate the
comparisons of sensitivities at higher model levels within the PBL and free troposphere by
showing consistent results to Fig. 6. This confirms the consistency in the impacts of emissions
to modeled atmospheric compositions between the forward and backward simulations, which
is the prerequisite for more detailed evaluations in the following Sections.
**3.2 Observing system simulation experiments with pseudo-CO observations**
Here we further evaluate the performance of GC-Adjoint-HEMCO in 4D-var
assimilations. OSSE is a useful method and has been widely used to evaluate the performance
of various data assimilation systems (Jones et al., 2003; Barré et al., 2015; Shu et al., 2022). In
contrast to assimilations by assimilating actual atmospheric observations, pseudo-observations
are usually generated by model simulations and then assimilated in OSSE. The true
atmospheric states are known in OSSEs as they are used to produce the pseudo-observations,
and consequently, the difference between assimilated and true atmospheric states describes the
capability of the assimilation systems to converge to the true atmospheric states in assimilations
when assimilating actual observations.
The pseudo-observations in this work are produced by archiving CO concentrations from
GC-Adjoint-HEMCO forward simulations with the CO emissions unchanged (i.e., the default
CO emission inventory such as CEDS, MIX and NEI2011). According to the usage of pseudo-
observations, two types of OSSE are performed in this work: 1) full modeled CO fields are
assimilated as pseudo-observations so that we have pseudo-CO observations at every grid/level
and time step (hereafter referred to as OSSE-FullOBS). This experiment is designed to evaluate
the performance of the assimilation system under ideal conditions with full coverage of
observations. 2) The modeled CO fields are sampled at the locations/times of MOPITT CO
observations and smoothed with MOPITT a priori concentrations and averaging kernels to
produce MOPITT-like pseudo-CO observations (hereafter referred to as OSSE-MOPITT). This
experiment is designed to evaluate the performance of the assimilation system under actual
conditions with limited coverage of observations.
In the inverse analysis with the pseudo-CO observations, we reduce the anthropogenic
CO emissions by 50% so that the objective of the OSSE is to produce scaling factors that can
return the source estimate to the default emissions (i.e., scaling factors of 1.0). Fig. 7A shows
the annual scaling factors in 2015 in OSSE-FullOBS. After 40 iterations, the a posteriori
anthropogenic CO emission estimates converge to the true states in all major emission regions.
As shown in Table 3, the regional scaling factors of OSSE-FullOBS are 1.00, 0.97, 0.97, 1.00,
0.98 and 0.94 for anthropogenic CO emissions over Asia, North America, Africa, South
America, Europe, and Australia, respectively.
Furthermore, Fig. 7D shows the annual scaling factors in OSSE-MOPITT, which are
noticeably worse than those in Fig. 7A. The regional scaling factors of OSSE-MOPITT are
1.04, 0.88, 1.01, 1.02, 0.84 and 0.81 for anthropogenic CO emissions over Asia, North
America, Africa, South America, Europe, and Australia, respectively. With respect to OSSE-
FullOBS, the limited coverage of observations in OSSE-MOPITT has resulted in
approximately 15% underestimations in the a posteriori CO emission estimates over North
America and Europe. In addition, Fig. 7B-C and Fig. 7E-F show the a priori and a posteriori
biases in the modeled CO columns. We find dramatic improvements in the modeled CO
columns, which confirms the reliability of the 4D-var assimilation system. The difference
between Fig. 7B and 6E reflects the influence of the application of MOPITT averaging kernels,
which lead to larger negative biases in the a priori simulation. It should be noted that we cannot
expect comparable improvement in the actual assimilations because of the potential effects of
model and observation errors.

**3.3 Anthropogenic CO emissions constrained with MOPITT CO observations**

As an example of the application of GC-Adjoint-HEMCO, here we constrain
anthropogenic CO emissions in 2015 by assimilating MOPITT CO observations. Fig.8A shows
the relative differences between modeled and MOPITT CO columns at the beginning of each
month in 2015 (i.e., biases in monthly initial CO conditions) in the original GEOS-Chem
simulations. We find dramatic underestimations in the modeled CO columns by approximately
30-40%. As indicated by previous studies (Jiang et al., 2013; Jiang et al., 2017), the biases in
monthly initial CO conditions are caused by model biases in CO concentrations accumulated
in previous months. Considering that the lifetime of CO is approximately 2-3 months, the
negative biases in the initial conditions can result in negative biases in the modeled CO
concentration in the following month. A lack of consideration of these biases, as shown in Fig.
8A, can thus result in overestimations in the derived monthly CO emission estimates because
the assimilation system will tend to adjust emissions to reduce the initial condition-induced
biases.
Following Jiang et al. (2017), a suboptimal sequential Kalman filter (Todling and Cohn,
1994; Tang et al., 2022) was employed in this work to optimize the modeled CO concentrations
with an hourly resolution by combining GC-Adjoint-HEMCO forward simulation and
MOPITT CO observations. The CO concentrations provided by the Kalman filter assimilations
were archived at the beginning of each month, which were used as the optimized monthly initial
CO conditions in the inverse analysis. As shown in Fig. 8B, the biases in the modeled CO
columns in the optimized initial CO conditions are pronounced lower than those in the original
simulation (Fig. 8A). The optimization of the initial CO conditions is essential for our inverse
analysis, as it can ensure that the adjustments in CO emissions are dominated by the differences
between simulations and observations in the current month instead of the 30-40%
underestimations in CO columns accumulated in previous months.
Fig. 9A shows the distribution of a priori anthropogenic CO emissions in 2015. The
regional a priori anthropogenic CO emissions (as shown in Table 4) are 243.53, 34.42, 23.24,
30.39, 25.94 and 2.02 Tg/y over Asia, North America, Africa, South America, Europe, and
Australia, respectively. As shown in Fig. 9B, our inverse analysis suggests a wide distribution
of underestimations in the a priori anthropogenic CO emissions in 2015 except in E. China.
The regional scaling factors (Table 4) are 1.16, 1.47, 1.52, 1.41, 1.60 and 1.38, and the a
posteriori anthropogenic CO emissions are 283.20, 50.47, 35.34, 42.92, 41.62 and 2.79 Tg/y
over Asia, North America, Africa, South America, Europe, and Australia, respectively. As
shown in Fig. 9C, we find noticeable underestimations in the modeled CO columns in the a
priori simulations, despite the negative biases being much weaker than those in Fig. 8A due to
the optimization of the initial CO conditions. The negative biases are effectively reduced in the
a posteriori simulation driven by the a posteriori CO emission estimates (Fig. 9D).
Finally, we compare the a posteriori CO emission estimates in this work with Jiang et al.
(2017), who constrained CO emissions in 2001-2015 with GC-Adjoint-STD by assimilating
the same MOPITT CO observations. As shown in Table 4, the a posteriori anthropogenic CO
emission estimates in this work match well with Jiang et al. (2017) in North America and Africa
but are 38%, 157% and 228% higher than those in Jiang et al. (2017) in Asia, South America
and Australia, respectively. A major discrepancy between this work and Jiang et al. (2017) is
the treatment of ocean grids. Jiang et al. (2017) defined ocean grids as continental boundary
conditions, which were rewritten hourly using the optimized CO concentrations archived from
the suboptimal sequential Kalman filter by assimilating MOPITT CO observations. Only
MOPITT data over land were assimilated in the 4D-var assimilations in Jiang et al. (2017)
without any change in CO distribution over the ocean. In addition, the large differences in
chemical sources and sinks between GC-Adjoint-HEMCO and GC-Adjoint-STD, for example,
lower VOC-generated CO emissions by 40-60% and higher CO sinks by 20-40% in GC-
Adjoint-HEMCO, as shown in Table 2, may also contribute to the discrepancy in the derived
a posteriori CO emission estimates.

As shown in Fig. 9D, the a posteriori simulation demonstrates positive biases in CO

columns over China and Southeast Asia, which is a signal of overestimated local CO emissions;
meanwhile, the negative biases over the northern Pacific Ocean are reduced in the a posteriori
simulation. The negative biases over the remote ocean are more affected by CO chemical
sources and sinks; however, biases in chemical sources cannot be effectively adjusted because
of the global uniform scaling factor for $CH_4$-generated CO emissions; biases in chemical sinks
cannot be adjusted because of the fixed OH fields in the tagged-CO simulation. Jiang et al.
(2017) tried to address this problem by defining continental boundary conditions so that the
inverse analysis is dominated by local MOPITT observations to avoid the influence of model
biases accumulated within the long-range transport. Conversely, CO emissions over China and
Southeast Asia are overestimated in this work to offset the negative biases over the northern
Pacific Ocean. We expect similar overestimations in the a posteriori CO emission estimates
over South America, southern Africa, and Australia in this work because it is the effective
pathway to reduce the negative bias over the ocean in the Southern Hemisphere.

**4. Conclusion**

This work demonstrates our efforts on the development of a new framework to facilitate

emission inventory updates in the adjoint of GEOS-Chem model. The major advantage of this
new framework is good readability and extensibility, which allows us to conveniently support
HEMCO emission inventories, including CEDS, MIX, NEI2011, DICE_AF, AF_EDGAR43,
APEI and GFED4. The updated emission inventories are critical for reliable sensitivity
analyses, as well as better convergence of assimilations by setting a more reasonable a priori
penalty in the cost function. Second, we developed new modules to support MERRA-2
meteorological data, which allows us to perform long-term inverse analysis with consistent
meteorological data in 1979-present. We evaluated the performances of the developed
capabilities by validating the diagnostic outputs of CO emissions, modeled surface and column
CO concentrations in forward simulations, and adjoint gradients of global CO concentrations
to CO emissions with respect to the finite difference gradients.

Two types of OSSE were conducted to evaluate the model performance in 4D-var

assimilations. The a posteriori CO emissions converged to the true states in all major emission
regions with fully covered pseudo-CO observations; the limited coverage of observations by
sampling the pseudo-CO observations at the locations/times of MOPITT CO observations and
smoothing with MOPITT averaging kernels resulted in approximately 15% underestimations
in the a posteriori CO emissions over North America and Europe. Furthermore, as an example
application of the developed capabilities, we constrain anthropogenic CO emissions in 2015
by assimilating MOPITT CO observations. The a posteriori anthropogenic CO emission
estimates derived in this work match well with Jiang et al. (2017) in North America and Africa
but are overestimated in Asia, South America and Australia, which could be associated with
the different treatment of MOPITT CO observations over ocean grids and the large differences
in CO chemical sources and sinks. The capabilities developed in this work are a useful
extension for the adjoint of GEOS-Chem model. More efforts are needed to support emissions
inventories associated with full chemistry simulations, as well as integration of these
capabilities with the standard GEOS-Chem adjoint code base for better development of the
community of the adjoint of GEOS-Chem model.

**Code and data availability:** The MOPITT CO data can be downloaded from
https://asdc.larc.nasa.gov/data/MOPITT/. The GEOS-Chem model (version 12.8.1) can be
downloaded from http://wiki.seas.harvard.edu/geos-chem/index.php/GEOS-Chem_12#12.8.1.
The adjoint of GEOS-Chem model (GC-Adjoint-STD) can be downloaded from
http://wiki.seas.harvard.edu/geos-chem/index.php/GEOS-Chem_Adjoint. The adjoint of
GEOS-Chem model (GC-Adjoint-HEMCO) can be downloaded from
https://doi.org/10.5281/zenodo.7512111.

**Author Contributions**: Z.J. designed the research. Z.T. developed the model code and
performed the research. Z.J. and Z.T. wrote the manuscript. All authors contributed to
discussions and editing the manuscript.

**Competing interests**: The authors declare that they have no conflicts of interest.

**Acknowledgments:** We thank the providers of the MOPITT CO data. The numerical
calculations in this paper have been done on the supercomputing system in the Supercomputing
Center of University of Science and Technology of China. This work was supported by the
Hundred Talents Program of Chinese Academy of Science and National Natural Science
Foundation of China (42277082, 41721002).

**Tables and Figures**
**Table 1.** CO emissions for each inventory in 2015 with unit Tg/y.

**Table 2.** Regional combustion CO emissions, VOC-generated CO (PCO_NMVOC), $CH_4$-
generated CO (PCO_CH4), CO sinks (CO_OH, calculated as CO_OH = KRATE×CO×OH),
and simulated surface and column CO concentrations in 2015. The region definitions are shown
in Fig. 2A.

**Table 3.** Annual scaling factors of anthropogenic CO emissions in OSSEs. The scaling factors represent the ratio of the estimated to true emissions. The ratio for the first guess is 0.5. The actual value is 1.0. The pseudo-observations are produced by GC-Adjoint-HEMCO forward simulation. The full modeled CO fields are used in OSSE-FullOBS as pseudo-CO observations. The modeled CO fields are smoothed with MOPITT averaging kernels to produce MOPITT-like pseudo-CO observations in OSSE-MOPITT.

**Table 4.** Regional anthropogenic CO emissions (with unit Tg/y) and annual scaling factors in 2015 in this work and Jiang et al. 2017.

**Fig. 1.** Framework to read the updated emission inventories in GC-Adjoint-HEMCO.

**Fig. 2.** Total combustion CO emissions in 2015 from (a) GC-v12; (b) GC-Adjoint-HEMCO; (c) GC-Adjoint-STD. The unit is molec/cm$^2$/s.

**Fig. 3.** Monthly variation in combustion CO emissions in 2015 from GC-v12, GC-Adjoint-HEMCO and GC-Adjoint-STD.

**Fig. 4.** Averages of surface CO concentrations (unit ppbv) in 2015 from (a) GC-Adjoint-HEMCO driven by MERRA-2, (b) GC-Adjoint-HEMCO driven by GEOS-FP and (c) their difference; (d-f) same as panels a-c, but for CO columns (column-averaged dry-air mole fractions, Xco).

**Fig. 5.** Averages of surface CO concentrations (unit ppbv) in 2015 from (a) GC-v12; (b) GC-Adjoint-HEMCO; (c) GC-Adjoint-STD; (d-f) same as panels a-c, but for CO columns (column-averaged dry-air mole fractions, Xco).

**Fig. 6.** Comparison of sensitivities of global CO concentrations (LFD_GLOB and model level 1) to CO emission scaling factors calculated using the adjoint method vs. the finite difference method. (a-c) the effects of convection, PBL mixing and advection with 24-hour assimilation window; (d-f) the combined effects (the advection process is turned off) with increased assimilation windows.

Fig. 7. (a) Annual scaling factors in OSSE-FullOBS. The scaling factors represent the ratio of the estimated to true emissions. The ratio for the first guess is 0.5. The actual value is 1.0. (b-c) the a priori and a posteriori biases calculated by (model-observation)/observation in OSSE-Full. (d-f) same as panels a-c, but for OSSE-MOPITT.

Fig. 8. (a) Biases in monthly initial CO conditions in 2015 in the original GEOS-Chem simulation. (b) same as panel a, but with optimized initial CO conditions provided by suboptimal sequential Kalman filter. The biases are calculated by (model-MOPITT)/MOPITT.

Fig. 9. (a) A priori anthropogenic CO emissions in 2015 with unit molec/cm$^2$/s; (b) Annual scaling factors for CO emissions in 2015. The scaling factors represent the ratio of the estimated to true emissions. (c-d) the a priori and a posteriori biases calculated by (model-MOPITT)/MOPITT.

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

| Inventories | GC-v12 | GC-Adjoint-HEMCO | Inventories | GC-Adjoint-STD |
|---|---|---|---|---|
| CEDS | 613.57 | 613.85 | GEIA | 445.88 |
| MIX | 321.18 | 321.71 | INTEX-B | 353.03 |
| NEI2011 | 35.83 | 37.70 | NEI2008 | 52.87 |
| DICE_AF + AF_EDGAR43 | 83.42 | 83.02 | \ | \ |
| APEI | 6.10 | 6.17 | CAC | 10.20 |
| GFED4 | 437.13 | 435.89 | GFED3 | 382.04 |

**Table 1.** CO emissions for each inventory in 2015 with unit Tg/y.

| Region / Version | Combustion Emission (Tg/y) | | | PCO_NMVOC (kg/s) | | | PCO_CH4 (kg/s) | | |
|---|---|---|---|---|---|---|---|---|---|
| | GC-v12 | GC-Adjoint-HEMCO | GC-Adjoint-STD | GC-v12 | GC-Adjoint-HEMCO | GC-Adjoint-STD | GC-v12 | GC-Adjoint-HEMCO | GC-Adjoint-STD |
| Asia | 320.66 | 320.38 | 331.65 | 15.49 | 15.52 | 22.37 | 14.21 | 14.40 | 10.67 |
| North America | 73.96 | 66.93 | 60.65 | 7.05 | 6.83 | 14.75 | 7.45 | 7.66 | 5.23 |
| Africa | 199.51 | 193.29 | 179.22 | 34.57 | 33.92 | 52.38 | 19.57 | 19.85 | 16.18 |
| South America | 79.04 | 78.91 | 75.82 | 44.15 | 42.55 | 74.64 | 17.14 | 17.42 | 14.08 |
| Europe | 31.58 | 30.96 | 48.48 | 4.20 | 4.14 | 10.17 | 7.13 | 7.41 | 4.58 |
| Australia | 12.24 | 11.99 | 22.87 | 21.23 | 20.68 | 48.89 | 13.88 | 14.62 | 10.67 |
| Region / Version | CO_OH (kg/s) | | | CO (surface ppbv) | | | CO (column xco) | | |
| | GC-v12 | GC-Adjoint-HEMCO | GC-Adjoint-STD | GC-v12 | GC-Adjoint-HEMCO | GC-Adjoint-STD | GC-v12 | GC-Adjoint-HEMCO | GC-Adjoint-STD |
| Asia | 52.26 | 51.34 | 40.87 | 179.56 | 184.29 | 187.90 | 90.23 | 88.16 | 89.58 |
| North America | 23.02 | 22.57 | 16.20 | 120.38 | 113.49 | 108.27 | 79.16 | 76.27 | 71.35 |
| Africa | 63.78 | 61.84 | 51.03 | 133.56 | 127.38 | 141.97 | 84.26 | 81.52 | 86.36 |
| South America | 49.06 | 48.85 | 41.25 | 107.98 | 106.16 | 132.24 | 72.93 | 70.67 | 78.75 |
| Europe | 20.65 | 20.92 | 14.27 | 112.88 | 111.33 | 120.09 | 74.83 | 72.34 | 70.45 |
| Australia | 31.42 | 31.98 | 25.27 | 67.45 | 65.00 | 84.80 | 56.35 | 54.02 | 61.15 |

**Table 2.** Regional combustion CO emissions, VOC-generated CO (PCO_NMVOC), $CH_4$-generated CO (PCO_CH4), CO sinks (CO_OH, calculated as CO_OH = KRATE×CO×OH), and simulated surface and column CO concentrations in 2015. The region definitions are shown in Fig. 2A.

|  | Scaling Factors OSSE-FullOBS | Scaling Factors OSSE-MOPITT |
|---|---|---|
| Asia | 1.00 | 1.04 |
| North America | 0.97 | 0.88 |
| Africa | 0.97 | 1.01 |
| South America | 1.00 | 1.02 |
| Europe | 0.98 | 0.84 |
| Australia | 0.94 | 0.81 |

**Table 3.** Annual scaling factors of anthropogenic CO emissions in OSSEs. The scaling factors represent the ratio of the estimated to true emissions. The ratio for the first guess is 0.5. The actual value is 1.0. The pseudo-observations are produced by GC-Adjoint-HEMCO forward simulation. The full modeled CO fields are used in OSSE-FullOBS as pseudo-CO observations. The modeled CO fields are smoothed with MOPITT averaging kernels to produce MOPITT-like pseudo-CO observations in OSSE-MOPITT.

|  |  | Asia | North America | Africa | South America | Europe | Australia |
|---|---|---|---|---|---|---|---|
| **This work** | A priori CO emissions | 243.53 | 34.42 | 23.24 | 30.39 | 25.94 | 2.02 |
|  | A posteriori CO emissions | 283.20 | 50.47 | 35.34 | 42.92 | 41.62 | 2.79 |
|  | Scaling Factors | 1.16 | 1.47 | 1.52 | 1.41 | 1.60 | 1.38 |
| **Jiang et al. 2017** | A priori CO emissions | 270.50 | 43.70 | 29.39 | 17.47 | 44.45 | 0.83 |
|  | A posteriori CO emissions | 205.40 | 47.06 | 35.04 | 16.67 | 53.58 | 0.82 |
|  | Scaling Factors | 0.76 | 1.08 | 1.19 | 0.95 | 1.21 | 0.99 |

**Table 4.** Regional anthropogenic CO emissions (with unit Tg/y) and annual scaling factors in 2015 in this work and Jiang et al. 2017.

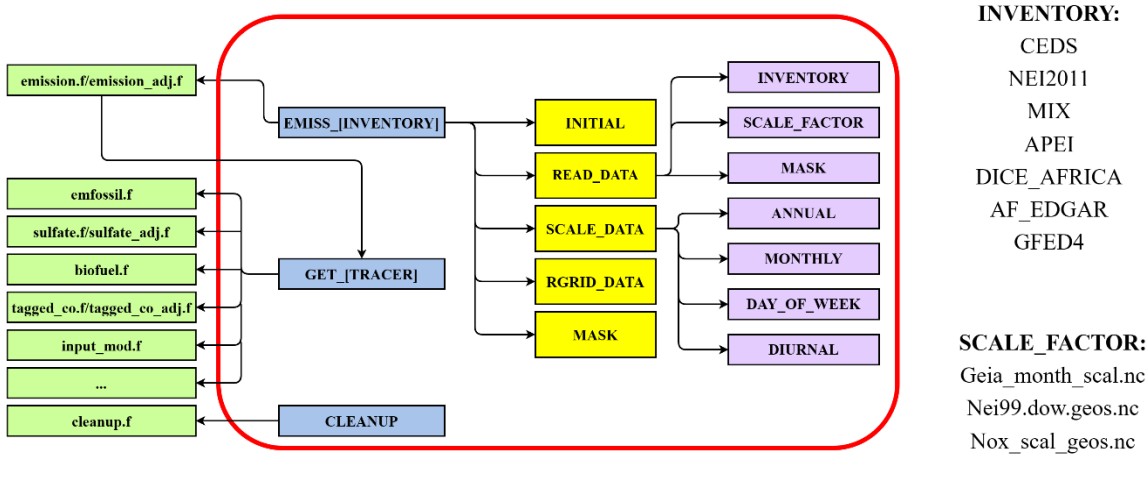

**Fig. 1.** Framework to read the updated emission inventories in GC-Adjoint-HEMCO.

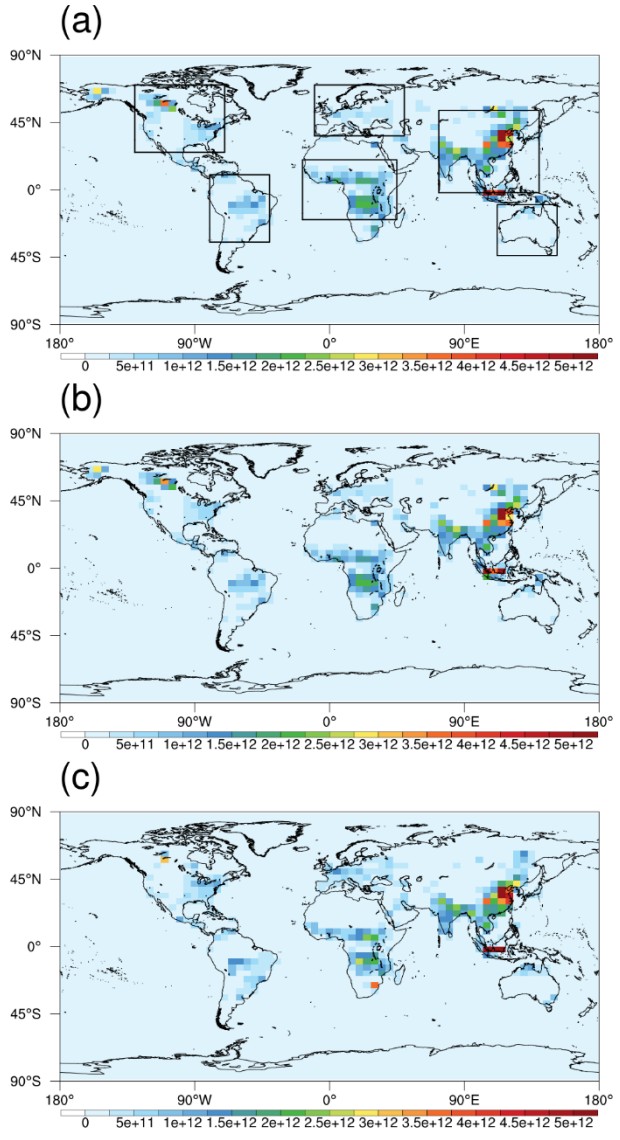

**Fig. 2.** Total combustion CO emissions in 2015 from (a) GC-v12; (b) GC-Adjoint-HEMCO; (c) GC-Adjoint-STD. The unit is molec/cm$^2$/s.

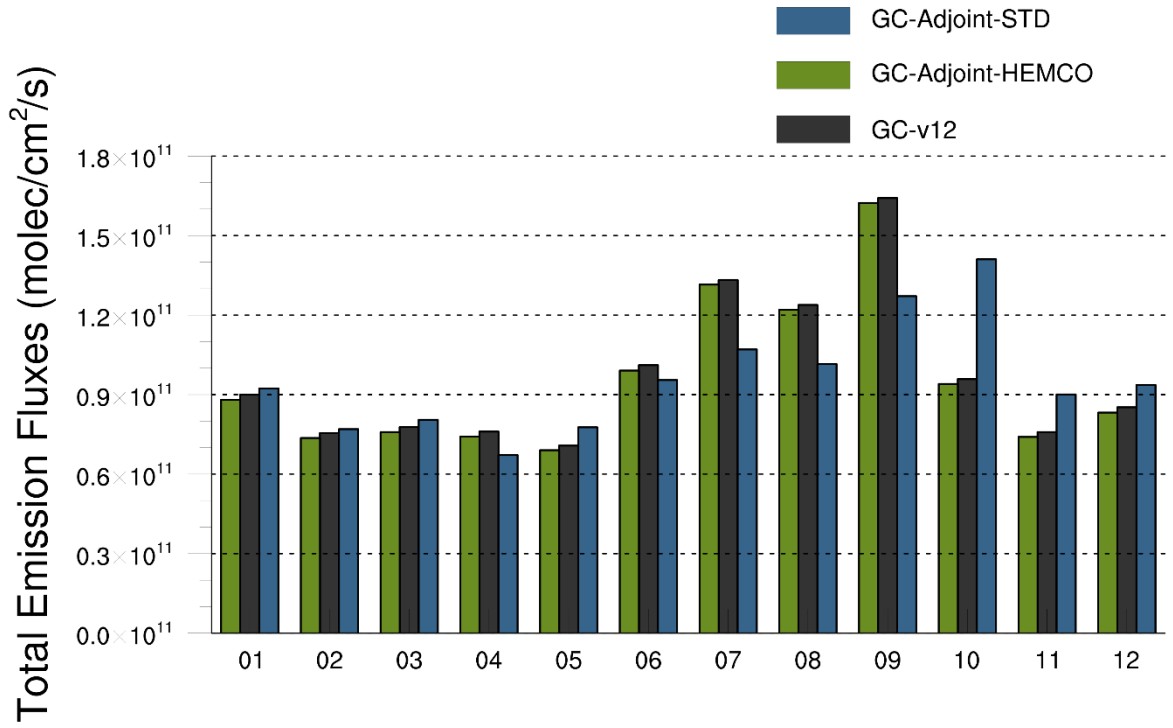

**Fig. 3.** Monthly variation in combustion CO emissions in 2015 from GC-v12, GC-Adjoint-HEMCO and GC-Adjoint-STD.

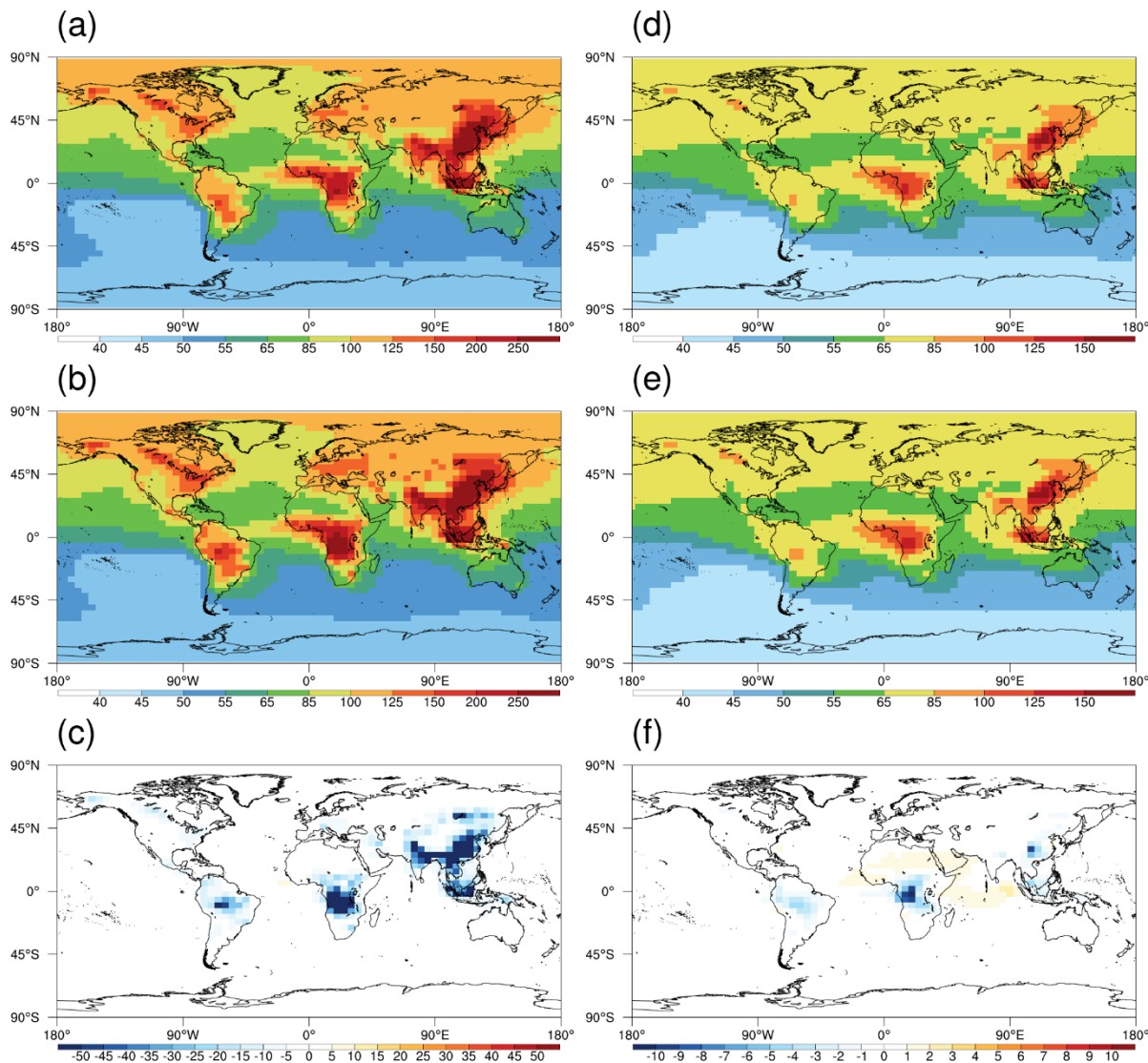

**Fig. 4.** Averages of surface CO concentrations (unit ppbv) in 2015 from (a) GC-Adjoint-HEMCO driven by MERRA-2, (b) GC-Adjoint-HEMCO driven by GEOS-FP and (c) their difference; (d-f) same as panels a-c, but for CO columns (column-averaged dry-air mole fractions, Xco).

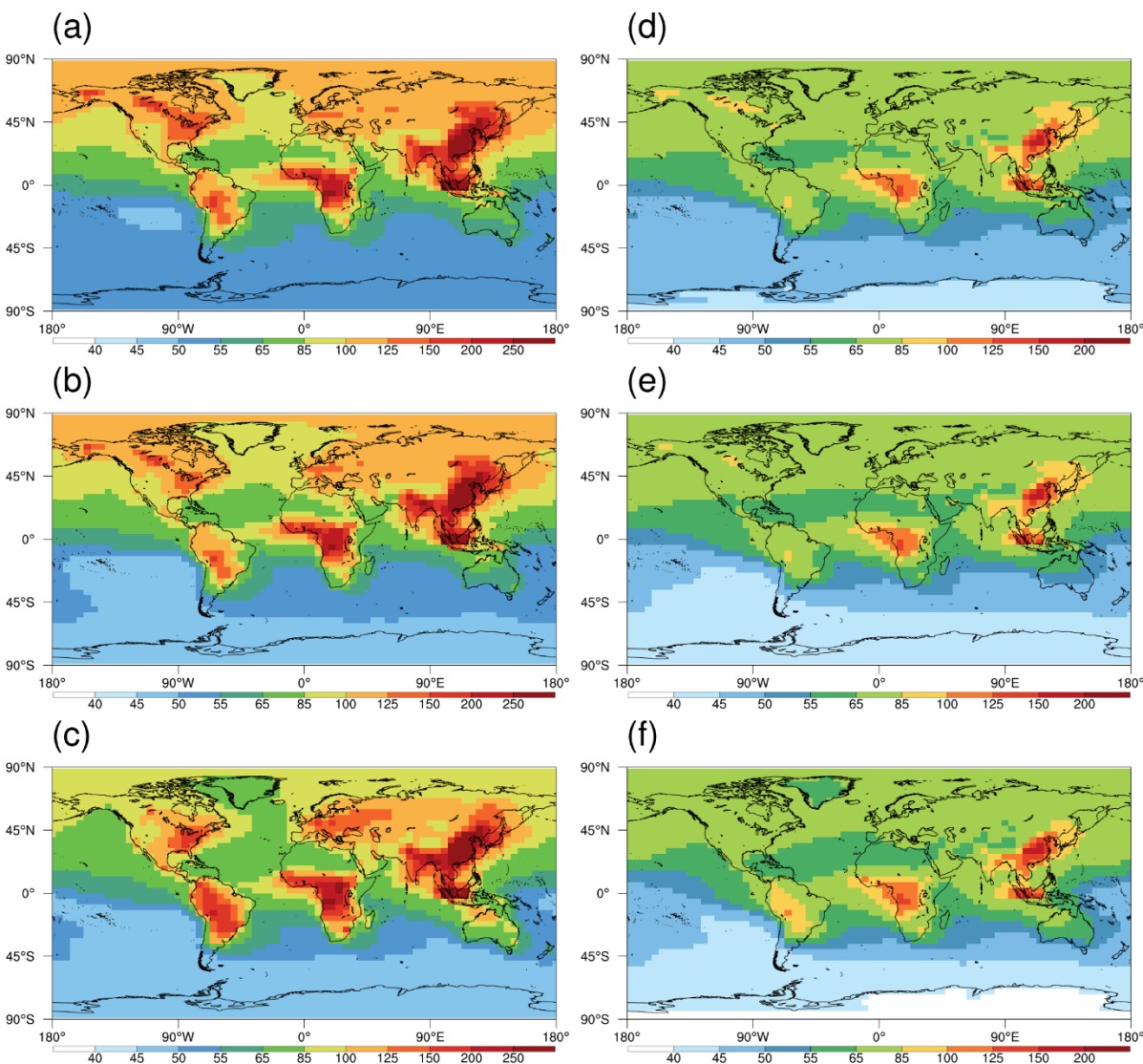

**Fig. 5**. Averages of surface CO concentrations (unit ppbv) in 2015 from (a) GC-v12; (b) GC-Adjoint-HEMCO; (c) GC-Adjoint-STD; (d-f) same as panels a-c, but for CO columns (column-averaged dry-air mole fractions, Xco).

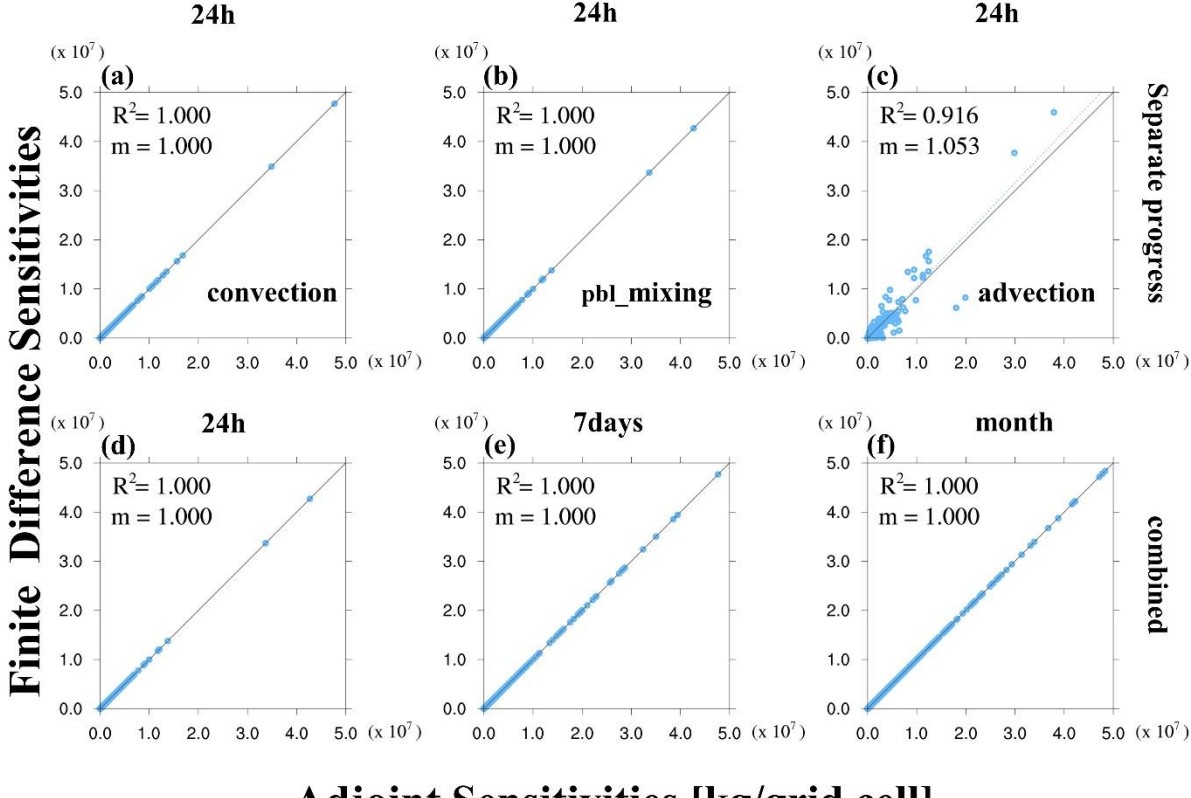

**Fig. 6**. Comparison of sensitivities of global CO concentrations (LFD_GLOB and model level 1) to CO emission scaling factors calculated using the adjoint method vs. the finite difference method. (a-c) the effects of convection, PBL mixing and advection with 24-hour assimilation window; (d-f) the combined effects (the advection process is turned off) with increased assimilation windows.

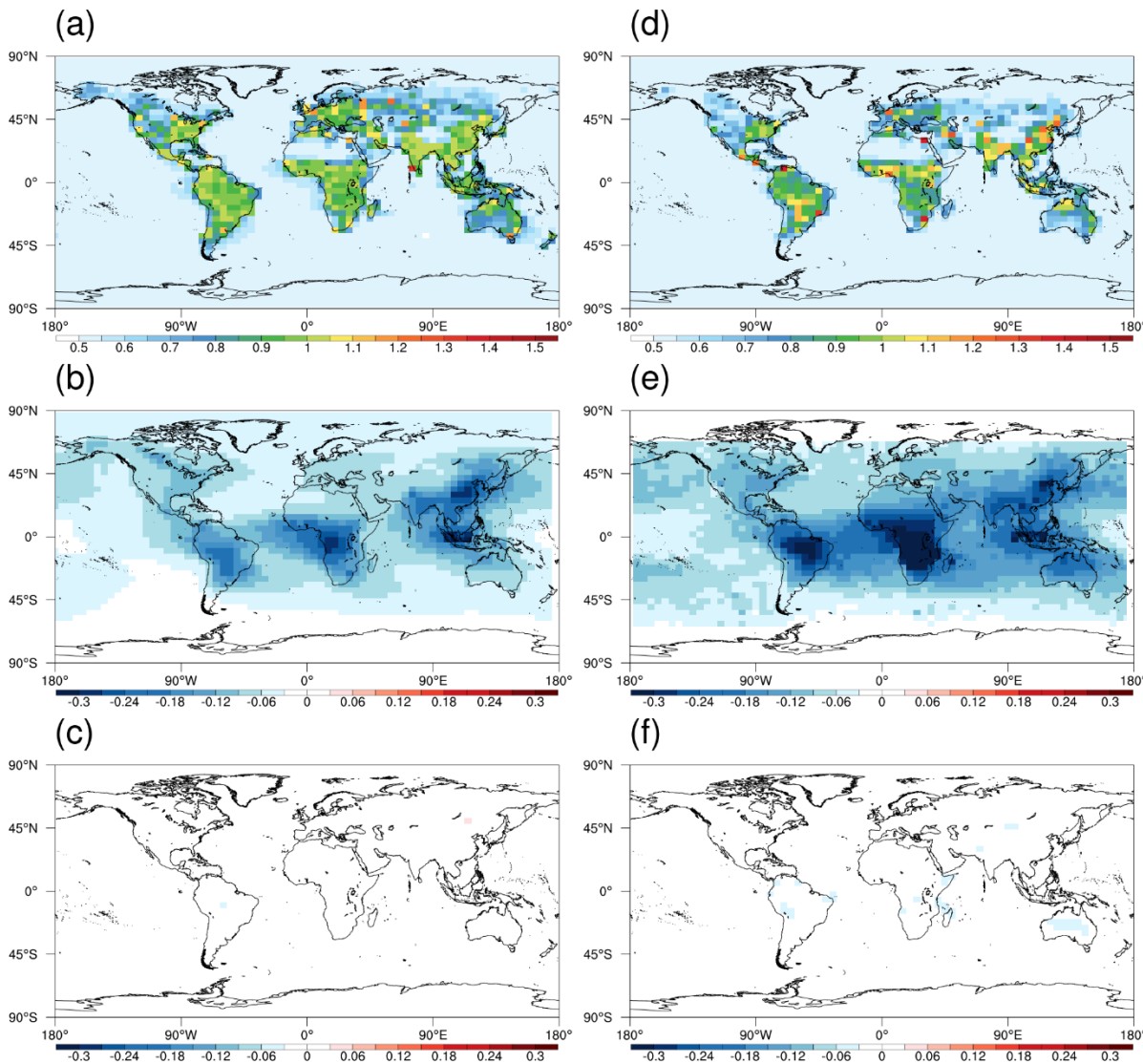

**Fig. 7**. (a) Annual scaling factors in OSSE-FullOBS. The scaling factors represent the ratio of the estimated to true emissions. The ratio for the first guess is 0.5. The actual value is 1.0. (b-c) the a priori and a posteriori biases calculated by (model-observation)/observation in OSSE-Full. (d-f) same as panels a-c, but for OSSE-MOPITT.

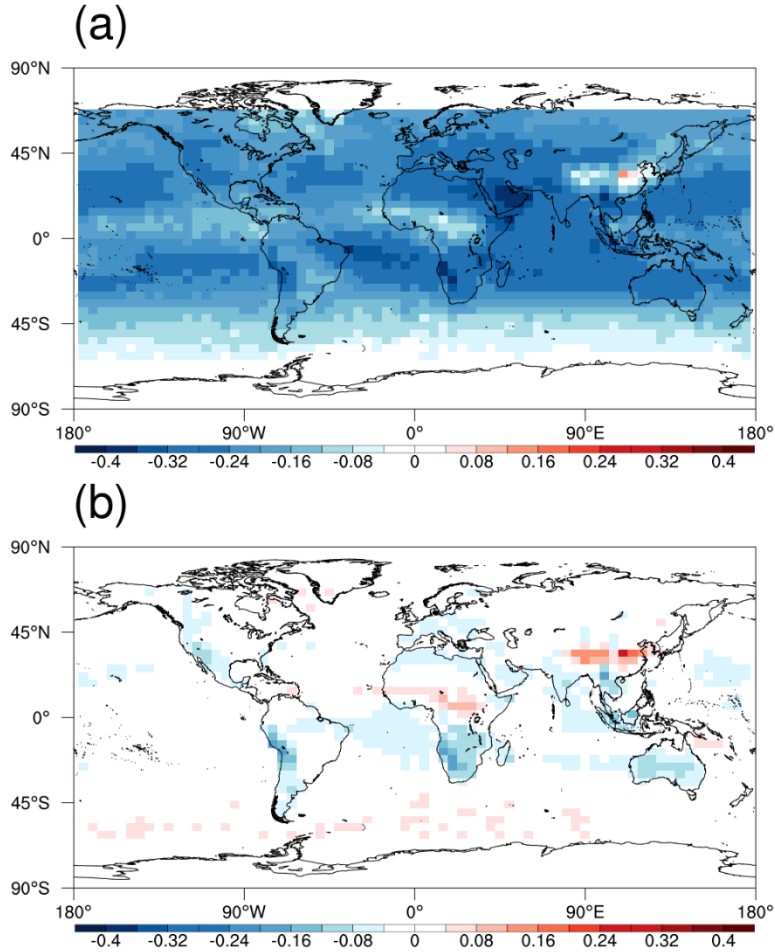

**Fig. 8**. (a) Biases in monthly initial CO conditions in 2015 in the original GEOS-Chem simulation. (b) same as panel a, but with optimized initial CO conditions provided by suboptimal sequential Kalman filter. The biases are calculated by (model-MOPITT)/MOPITT.

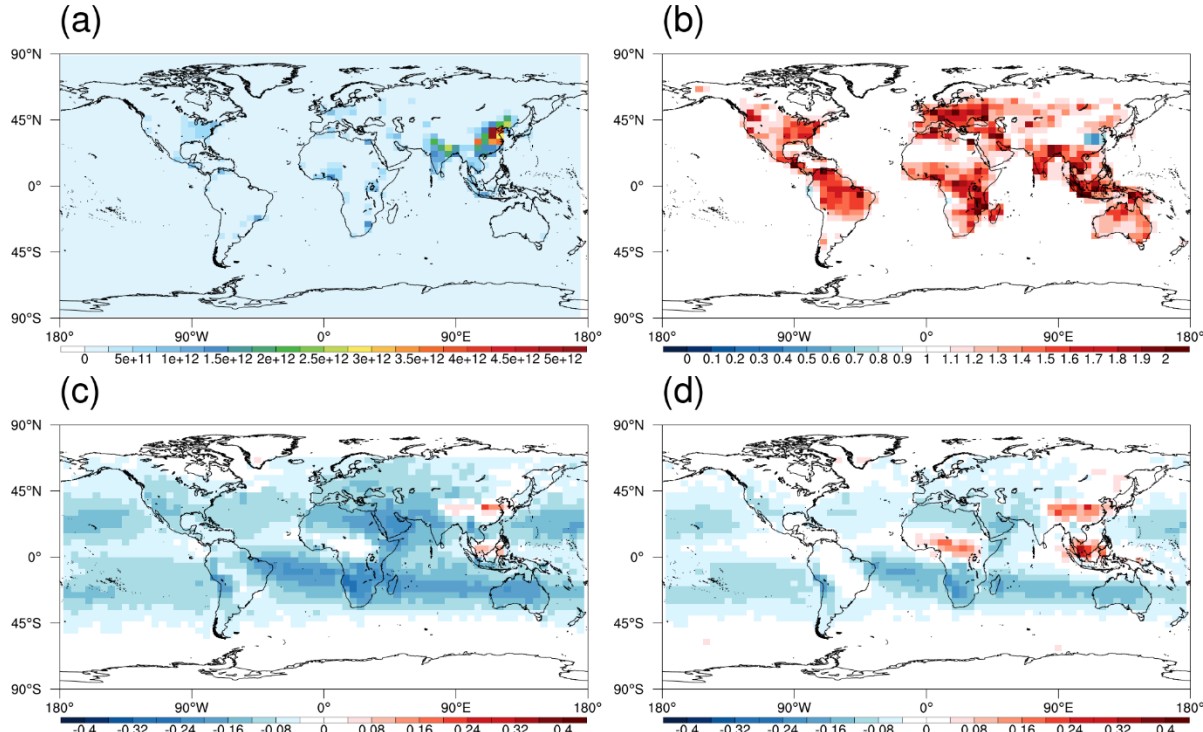

**Fig. 9**. (a) A priori anthropogenic CO emissions in 2015 with unit molec/cm$^2$/s; (b) Annual scaling factors for CO emissions in 2015. The scaling factors represent the ratio of the estimated to true emissions. (c-d) the a priori and a posteriori biases calculated by (model-MOPITT)/MOPITT.