# Peer review of "The capabilities of the adjoint of GEOS-Chem model to support HEMCO emission inventories and MERRA-2 meteorological data"

_Geoscientific Model Development, 2023_

## Author Response (AR2)

We thank the editor and reviewers for their thoughtful and detailed comments. We have revised this manuscript carefully based on the comments. The text in the response file which is quoted from the revised manuscript is distinguished in red font *with Line numbers*. The corresponding text is also highlighted in the tracked change files of the revised manuscript. In addition, the text in the responses to reviewers has been adjusted to provide clearer answers to reviewers' questions, and the difference can be easily checked at the end of this file.

Below we respond to the individual comments.

Additional private note (visible to authors and reviewers only):

**Question**: Changes to the manuscript are not easily understood. For example, according to the track changes document, equation 1 now has a gamma parameter but equation 1 is not reproduced in the response to reviewers. Equation 2 may be new. Some text removed from the main text appears partially hidden/inaccessible in the track changes document. In addition, some revisions to the main text were not indicated as revisions in the response to reviewers. To proceed with review, please submit a revised author response where changes to the main text in response to reviewer comments are clearly indicated as revisions. Please use different text (bold or quotes or a different color or indentation) to show revisions to the manuscript in the response to reviewers. Please itemize and explain any changes to the main text that were not in response to reviewers and make sure the old and new text is easily determined from the track changes.

**Answer**: An important change in the revised manuscript is the evaluation of the reliability of backward simulations by comparing adjoint gradients with finite difference gradients as shown in Section 3.1. To help the readers understand the simplified adjoint gradients for sensitivity analysis (Eq. 3 in Section 3.1), the full adjoint gradient was exhibited in the revised manuscript (Eq. 2 in Section 2.1). In addition, the definition of the cost function (Eq. 1 in Section 2.1) was improved by including the parameter "gamma", which is used in 4D-var assimilations to adjust the weights of observational and a priori terms of the cost function. We are sorry for this confusion!

As suggested by the editor, the text in the response file which is quoted from the revised manuscript is distinguished in red font with Line numbers. The corresponding text is also highlighted in the tracked change files of the revised manuscript. As shown in the tracked change files, most revised text in the Abstract, Introduction, Methodology and Result Sections are quoted in the responses to reviewers. The revised discussion in Section 3.1 is summarized and exhibited in the responses to reviewers. In addition, two versions of tracked change files of the revised manuscript were uploaded in this round by showing the revisions inline and in balloons, respectively. Showing revision in balloons demonstrates the revised text clearly, while showing revision inline allows us to check the deleted text in detail.

The text in the responses to reviewers has been adjusted to provide clearer answers to reviewers' questions, and the difference can be easily checked at the end of this file. Thank the editor for the valuable suggestions! Hope the revised response and tracked change files clarify the changes better.

Reviewer #1

The manuscript presents an updated version of the adjoint model of the GEOS-Chem chemical transport model. The update supports a new version of the GEOS assimilated meteorology (MERRA-2) and also supports the emission module (HEMCO). The new version of the GEOS-Chem adjoint is then applied to assimilate pseudo-observations and MOPITT observations to constrain anthropogenic CO emissions.

I agree that model developments are crucial and important updates shall be documented in journal papers. However, I feel this manuscript is organized in a simple way: it is not convincing that the new updates are sufficiently important and a GMD paper is needed to document them. Further, it does not demonstrate that the application of the new adjoint can generate new knowledge. I think that the current manuscript needs a major revision to address the two concerns.

**Answer**: Thank you for the comments! The manuscript has been revised based on the comments.

Specific comments:

**Question**: 1) Page 5, Section 2.2. Here it is not clear how the structure of the adjoint code has been updated. Only the way the model reads the emissions? How is that different from the previous adjoint (GC-Adjoint-STD)? As HEMCO has been implemented in the GEOS-Chem forward model, what are the extra efforts to implement it in the adjoint?

**Answer**: As clarified in the revised version: "HEMCO was included in the GEOS-Chem forward simulations in v10-01. HEMCO is responsible for inputs of meteorological and emission data with default support for emission inventories such as CEDS, MIX and NEI2011" (*Lines 114-117 in the tracked change file*). There are noticeable differences between HEMCO and the adjoint of GEOS-Chem model, as the latter is based on GEOS-Chem v8. "First, meteorological and emission data are read with individual modules in the adjoint of GEOS-Chem model. Second, the inputs of emission inventories are undertaken by different modules that were developed individually with significant discrepancies in the source code. In addition, the file format (e.g., binary punch in the adjoint of GEOS-Chem that is the format of older GEOS-Chem versions in contrast to netCDF in HEMCO), emission variables and the usage methods of emission variables (e.g., emission hierarchy, scaling factors and time slice) are inconsistent. These differences have posed a barrier to the application of new emission inventories in the adjoint of GEOS-Chem model" (*Lines 119-126 in the tracked change file*).

"The lack of support to the updated emission inventories can affect the applications of the adjoint of GEOS-Chem model. First, adjoint-based sensitivity analyses are obtained by the backward simulations of atmospheric compositions (i.e., adjoint tracers) and the combination of adjoint tracers with emissions. Out-of-date emission inventories can thus result in inaccurate estimation of the adjoint sensitivities. Second, while inverse analyses are constrained by atmospheric observations, the updated emission inventories are still critical because they are helpful for better convergence of 4D-var assimilations by setting a more reasonable a priori penalty in the cost function" (*Lines 127-144 in the tracked change file*).

"Ideally, people should consider porting the complete HEMCO to the adjoint of GEOS-Chem model to match the new features in GEOS-Chem forward simulations. However, a complete port of HEMCO implies replacing the input framework of the adjoint of GEOS-Chem model, as well as restructuring of HEMCO and the adjoint of GEOS-Chem model to address the compatibility issues, which is very challenging and may not be necessary because the meteorological modules still work well in the adjoint of GEOS-Chem model" (*Lines 149-154 in the tracked change file*).

"Consequently, a major objective of this work is to design a new framework to facilitate emission inventory updates in the adjoint of GEOS-Chem model" (*Lines 154-156 in the tracked change file*). This new framework is not HEMCO, and is different from the original emission inventory modules in the adjoint of GEOS-Chem. "The major advantage of this new framework is good readability and extensibility, which allows us to support HEMCO emission inventories conveniently and to easily add more emissions inventories following future updates in GEOS-Chem forward simulations" (*Lines 15-18 in the tracked change file*).

As indicated by the reviewer, the development in this work is not "Only the way the model reads the emissions". As a 4D-var assimilation system, it is important to ensure consistent emissions in both forward and backward simulations. We have made corresponding modifications to both forward and backward modules, and the reliability of the backward simulations was validated by comparing adjoint gradients of global CO concentrations to CO emissions with finite difference gradients. The capabilities developed in this work are thus reliable and important for better applications of the adjoint of GEOS-Chem model in the future.

**Question**: 2) Section 2.2. Here CO emissions of GC-v12 and GC-Adjoint-HEMCO are compared. Again it is not clear that comparisons of the emissions are sufficiently important to be viewed as a major development. We would expect differences when using different emission inventories, and similar (if not the same) values when using the same emission inventories (as in GC-v12 and GC-Adjoint-HEMCO).

**Answer**: As clarified in the revised version: "In addition to baseline emission data, there are critical factors that affect the usage of emission data in the models. Reading the emission data correctly thus does not necessarily mean using emission data correctly. For example, emission hierarchy is used to prioritize emission fields within the same emission category. Emissions of higher hierarchy overwrite lower hierarchy data. Regional emission inventories usually have a higher hierarchy within their mask boundaries. Scaling factors are used to adjust the baseline emissions with annual, season, month, week, and 24-hour temporal scales. Time slice selection is used to define the usage methods of the emission data outside the original temporal range; for instance, data can be interpreted as climatology and recycled once the end of the last time slice is reached or be only considered as long as the simulation time is within the time range. Furthermore, there are experience parameters applied in files such as emfossil.f and tagged_co.f, which may not be compatible with HEMCO emission inventories. Consequently, we must validate the integrated emissions carefully to ensure that the abovementioned factors have been correctly applied and to ensure that the calculated

emissions are reasonable for individual inventories and the combination of all inventories" (*Lines 278-292 in the tracked change file*).

Furthermore, it should be noted that the comparison of the emissions in Section 2.2 is only the first step of our model evaluations. As clarified in the revised version: "The performances of the developed capabilities were evaluated with the following steps: 1) diagnostic outputs of carbon monoxide (CO) sources and sinks to ensure the correct reading and use of emission inventories; 2) forward simulations to compare the modeled surface and column CO concentrations among various model versions; 3) backward simulations to compare adjoint gradients of global CO concentrations to CO emissions with finite difference gradients; and 4) observing system simulation experiments (OSSE) to evaluate the model performance in 4D variational (4D-var) assimilations" (*Lines 20-27 in the tracked change file*).

The development of the new capabilities in this work is challenging due to the building of an integrated system involving the development of new modules, modifications of existing modules, and usage of various emission data with complex control parameters. Consequently, we have spent great efforts on model evaluation because these evaluations are important to ensure the reliability of the developed capabilities, which is the prerequisite to submitting our update to the standard GEOS-Chem adjoint code base for wider usage by the community.

The discussion in Section 2.2 has been revised. Thank the reviewer for pointing out this issue!

**Question**: 3) Page 10, Section 2.4. Here the authors stated that supporting MERRA-2 is more direct as it can follow the GEOS-FP fields. So how important is the update and any demonstration of that?

**Answer**: The importance of MERRA-2 meteorological data is reflected in long-term analysis with consistent meteorological data. As the reviewer suggested, the manuscript has been revised to clarify this point: "The adjoint of GEOS-Chem model does not support MERRA-2, and thus, long-term analysis must combine different meteorological reanalysis data, such as GEOS-4 (1985-2007), GEOS-5 (2004-2012) and GEOS-FP (2012-present). For instance, Jiang et al. [2017] constrained global carbon monoxide (CO) emissions in 2001-2015, while the derived trends in CO emissions in *Jiang et al.* [2017] could be affected by the discontinuity among various versions of the meteorological data (i.e., GEOS-4 in 2001-2003, GEOS-5 in 2004-2012 and GEOS-FP in 2013-2015) and the lack of consistency in the model physics of GEOS-5" (*Lines 105-112 in the tracked change file*).

This sentence has been revised: "The code porting to support MERRA-2 follows the current framework of the adjoint of GEOS-Chem model … " (*Lines 395-397 in the tracked change file*).

Reference:

Jiang, Z., Worden, J. R., Worden, H., Deeter, M., Jones, D. B. A., Arellano, A. F., and Henze, D. K.: A 15-year record of CO emissions constrained by MOPITT CO observations, Atmos Chem Phys, 17, 4565-4583, 10.5194/acp-17-4565-2017, 2017.

**Question**: 4) Section 3. The section applied the new version of GEOS-Chem adjoint to constrain CO anthropogenic emissions following previous works of the authors. This appears only to show that the adjoint model is running, and does not provide any new scientific findings. Why shall we need to use the new adjoint? Shall we get the same results if we still use the old version (GC-Adjoint-STD) with the prior emissions updated?

**Answer**: As the reviewer indicated, the assimilation experiment in Section 3.3 is designed to show the usability of the developed capabilities, because it is difficult to demonstrate the advantage of GC-Adjoint-HEMCO by performing an assimilation experiment for a single year. In our ongoing project, we are planning to reproduce Jiang et al. [2017] by constraining global CO emissions in 2001-2022 with different observations and OH fields, which is expected to better demonstrate the advantage of the developed capabilities.

A major objective of this work is to design a new framework to facilitate emission inventory updates in the adjoint of GEOS-Chem model. This new framework is not HEMCO, and is different from the original emission inventory modules in the adjoint of GEOS-Chem. The capabilities developed in this work are actually similar to the reviewer's suggestion, i.e., "use the old version (GC-Adjoint-STD) with the prior emissions updated", because the designed new framework provides a convenient pathway to support updated emission inventories. Furthermore, it should be noted that we also developed new modules to support MERRA-2 meteorological data. This allows us to perform long-term analysis with consistent meteorological data in 1979-present, which is not supported by GC-Adjoint-STD.

Reference:

Jiang, Z., Worden, J. R., Worden, H., Deeter, M., Jones, D. B. A., Arellano, A. F., and Henze, D. K.: A 15-year record of CO emissions constrained by MOPITT CO observations, Atmos Chem Phys, 17, 4565-4583, 10.5194/acp-17-4565-2017, 2017.

**Question**: 5) Units are missing for Figures 4 and 5.

**Answer**: The units have been added.

Reviewer #2

The authors developed an updated version (GC-Adjoint-HEMCO) of the adjoint of the GEOS-Chem model to support MERRA-2 meteorological data and HEMCO emission inventories. Their analysis demonstrates good consistency in the forward simulations between GC-Adjoint-HEMCO and standard GEOS-Chem. The reliability of GC-Adjoint-HEMCO in 4D-Var assimilation is further evaluated through observing system simulation experiments (OSSEs). The authors should have spent great efforts on the system development and presented comprehensive results.

This paper is well written. The GC-Adjoint-HEMCO is an important contribution to the community of the adjoint of the GEOS-Chem model. I recommend the paper for publication after consideration of the points below.

**Answer**: Thank you for the comments! As the reviewer indicated, we have spent great efforts on the development of these capabilities. The developed capabilities will be submitted to the standard GEOS-Chem adjoint code base as a part of our contributions to the development of the community of the adjoint of GEOS-Chem model.

Comments:

**Question**: Lines 63-66:It is suggested to provide more discussion to clarify the advantages of the newer emission inventories, as it is the major motivation of the development of GC-Adjoint-HEMCO.

**Answer**: Thank you for this suggestion! As discussed in the revised manuscript: "The lack of support to the updated emission inventories can affect the applications of the adjoint of GEOS-Chem model. First, adjoint-based sensitivity analyses are obtained by the backward simulations of atmospheric compositions (i.e., adjoint tracers) and the combination of adjoint tracers with emissions. Out-of-date emission inventories can thus result in inaccurate estimation of the adjoint sensitivities. Second, while inverse analyses are constrained by atmospheric observations, the updated emission inventories are still critical because they are helpful for better convergence of 4D-var assimilations by setting a more reasonable a priori penalty in the cost function. For instance, the a priori biomass burning CO emissions (GFED3, van der Werf et al. (2010)) in Jiang et al. (2017) lack interannual variabilities later than 2011. In order to obtain reasonable convergence of biomass burning emissions, the a priori biomass burning emissions in September-November 2006 were applied to September-November 2015 over Indonesia in Jiang et al. (2017)" (*Lines 127-148 in the tracked change file*).

Furthermore, it should be noted that the major advantage of our new framework is good readability and extensibility, which not only allows us to support HEMCO emission inventories conveniently, but also allows us to add more emission inventories following future updates in GEOS-Chem forward simulations easily. It is thus important for better applications of the adjoint of GEOS-Chem model in the future.

References:

Jiang, Z., Worden, J. R., Worden, H., Deeter, M., Jones, D. B. A., Arellano, A. F., and Henze, D. K.: A 15-year record of CO emissions constrained by MOPITT CO observations, Atmos Chem Phys, 17, 4565-4583, 10.5194/acp-17-4565-2017, 2017.

van der Werf, G. R., Randerson, J. T., Giglio, L., Collatz, G. J., Mu, M., Kasibhatla, P. S., Morton, D. C., DeFries, R. S., Jin, Y., and van Leeuwen, T. T.: Global fire emissions and the contribution of deforestation, savanna, forest, agricultural, and peat fires (1997–2009), Atmos Chem Phys, 10, 11707-11735, 10.5194/acp-10-11707-2010, 2010.

**Question**: Lines 67-81: It is unclear whether GC-Adjoint-HEMCO can perform assimilations with the full chemistry mode.

**Answer**: As discussed in the revised manuscript: "The capabilities developed in this work are thus based on the tagged-CO mode, as it can effectively accelerate the model development process. More efforts are needed in the future to extend these capabilities to support emissions inventories associated with the full chemistry simulations" (*Lines 178-181 in the tracked change file*). We are sorry for this confusion!

**Question**: Lines 115-118:Please clarify the criteria for assimilation convergence. Why were 40 iterations performed here?

**Answer**: The discussion has been revised: "Following Jiang et al. (2017), we performed 40 iterations (forward + backward simulations) for each month, which usually produced 6-8 accepted iterations (i.e., successful line searches in the large-scale bound constrained optimization (L-BFGS-B, Zhu et al. (1997)) to reduce the cost functions and adjoint gradients. The a posteriori CO emission estimates were calculated based on the last accepted iteration, which usually corresponded to the iteration with the lowest cost function" (*Lines 242-247 in the tracked change file*).

References:

Jiang, Z., Worden, J. R., Worden, H., Deeter, M., Jones, D. B. A., Arellano, A. F., and Henze, D. K.: A 15-year record of CO emissions constrained by MOPITT CO observations, Atmos Chem Phys, 17, 4565-4583, 10.5194/acp-17-4565-2017, 2017.

Zhu, C., Byrd, R. H., Lu, P., and Nocedal, J.: Algorithm 778: L-BFGS-B: Fortran Subroutines for Large-Scale Bound Constrained Optimization, ACM Transactions on Mathematical Software, 23, 550-560, 10.1145/279232.279236, 1997.

**Question**: Line 262-271:The numbers of relative differences are listed in this paragraph. More discussions are suggested to clarify the importance of better consistency between GC-Adjoint-HEMCO and GC-v12.

**Answer**: As the reviewer suggested, the discussion has been revised: "The reasonable emissions in the diagnostic outputs in Section 2 do not necessarily mean the correct integration of emissions in the simulations" (*Lines 423-424 in the tracked change file*). Consequently, we further evaluate the performance of GC-Adjoint-HEMCO in forward simulations. "The agreement between GC-v12 and GC-Adjoint-HEMCO confirms the reliability of GC-Adjoint-HEMCO in forward simulations" (*Lines 438-439 in the tracked change file*).

**Question**: Lines 324-325:It would be better to provide an explanation as why a negative deviation in the initial conditions would lead to an overestimation in the emissions.

**Answer**: Thank you for this suggestion! The discussion has been revised: "the biases in monthly initial CO conditions are caused by model biases in CO concentrations accumulated in previous months. Considering that the lifetime of CO is approximately 2-3 months, the negative biases in the initial conditions can result in negative biases in the modeled CO concentration in the following month. A lack of consideration of these biases, as shown in Fig. 8A, can thus result in overestimations in the derived monthly CO emission estimates because the assimilation system will tend to adjust emissions to reduce the initial condition-induced biases" (*Lines 532-539 in the tracked change file*).

Minors:

**Question**: Line 157-185, it might be worth adding some information (area, seasonal/daily/hourly factors, ) of the different inventorories in Table 1. So the descriptions in the text could be more brief.

**Answer**: Changed.

**Question**: Line 160, mid-week -> weekday?

**Answer**: Changed.

Reviewer #3

This manuscript is presented as a description of a new model / modeling capabilities, in particular running the GEOS-Chem adjoint model using MERRA-2 meteorology, and with emissions processed by HEMCO. While these are valuable developments, and the application shown here with regards to CO emissions estimation is interesting on its own, overall the manuscript is unfortunately problematic for the following reasons.

**Answer**: Thank you for the comments! The manuscript has been revised based on the comments.

**Question**: First, the codes updates included here are alone not significant enough to warrant publication as a stand-alone modeling paper. This is not a new model, just an update to one that is widely used, and not a major update.

**Answer**: As the reviewer indicated, the update here is not a stand-alone model. A major objective of this work is to design a new framework to facilitate emission inventory updates in the widely used adjoint of GEOS-Chem model. It should be noted that this new framework allows direct usage of native HEMCO emission inventories, which is different from the usage of emission data processed by HEMCO, i.e., while experienced users can run HEMCO to output emission data and then develop modules by themselves to read HEMCO-output data into the adjoint of GEOS-Chem, this process is inconvenient and unfriendly, particularly, the HEMCO has to be rerun if there is any change in the inventory settings.

In contrast, the capability to direct usage of native HEMCO emission inventories in this update is much more convenient and friendly to users, which is important for better development of the community of the adjoint of GEOS-Chem model. As shown in the following Figure, the updated emission inventories (such as CEDS, MIX and NEI2011) can be easily selected with simple menu options in the updated input.geos file to keep the same user experience as the standard version of adjoint of GEOS-Chem model. The developed capabilities were clarified in the revised manuscript: "The major advantage of this new framework is good readability and extensibility, which allows us to support HEMCO emission inventories conveniently and to easily add more emissions inventories following future updates in GEOS-Chem forward simulations. Furthermore, we developed new modules to support MERRA-2 meteorological data, which allows us to perform long-term analysis with consistent meteorological data in 1979-present" (*Lines 15-20 in the tracked change file*).

Furthermore, as indicated by the reviewer in the next question, model evaluation is often more time-consuming than code development itself, particularly, the development of the new capabilities in this work is challenging due to the building of an integrated system involving the development of new modules, modifications of existing modules

and usage of various emission data with complex control parameters. As clarified in the revised version, we have spent great efforts on model evaluation: "The performances of the developed capabilities were evaluated with the following steps: 1) diagnostic outputs of carbon monoxide (CO) sources and sinks to ensure the correct reading and use of emission inventories; 2) forward simulations to compare the modeled surface and column CO concentrations among various model versions; 3) backward simulations to compare adjoint gradients of global CO concentrations to CO emissions with finite difference gradients; and 4) observing system simulation experiments (OSSE) to evaluate the model performance in 4D variational (4D-var) assimilations" (*Lines 20-27 in the tracked change file*). The capabilities developed in this work are thus reliable and important for better applications of the adjoint of GEOS-Chem model in the future.

```
%%% EMISSIONS MENU %%%  :
Turn on emissions?      : T
Emiss timestep (min)    : 20
Include anthro emiss?   : T
 => Scale to (1985-2005): -1
 => Use EMEP emissions? : F
 => Use BRAVO emissions?: F
 => Use EDGAR emissions?: F
 => Use CEDS emissions? : T
 => Use MIX  emiss?     : T
 => Use APEI emissions? : T
 => Use NEI2005 emiss?  : F
 => Use NEI2011 emiss?  : T
 => Use RETRO emiss?    : F
 => Use RCP   emiss?    : F
       RCP scenario ?   : RCP60
       RCP year ?       : 2010
 => Use AFRICA emiss?   : T
```

**Question**: Second, the authors only validate one aspect of their model updats — the forward model performance — but do not evaluate nor validate the adjoint code. This is a good sanity check, but not sufficient for a publication, nor for demonstration that their updates are correct and complete. As adjoint code developers well know, it is the verification of the adjoint sensitivities following any code update which is often more time consuming than the forward model update. However, the authors have not demonstrated that adjoint sensitivities are correctly propagated through HEMCO, or when running MERRA-2 meteorology. This needs to be demonstrated via numerical evaluation of the accuracy of their adjoint sensitivities for horizontal advection, vertical convection, and emissions scaling factors. This is a standard test for any update to the GEOS-Chem adjoint, and their code is not ready for submission to the community code repository until this has been demonstrated. If the updates are so trivial that the authors feel numerical tests are not required, this would further emphasize that the work, while commendable, is not substantial enough to warrant a stand-alone publication.

**Answer**: New Fig. 6 was added in the revised version to show the comparison of adjoint and finite difference gradients of global CO concentrations to CO emission, by turning on the convection, planetary boundary layer mixing and advection processes

individually, and the effects of combined model processes with various assimilation windows (turning off advection as suggested by Henze et al. (2007)). We find good agreement between the adjoint and finite difference gradients. This confirms the consistency between forward and backward simulations in the developed capabilities.

Reference:

Henze, D. K., Hakami, A., and Seinfeld, J. H.: Development of the adjoint of GEOS-Chem, Atmos Chem Phys, 7, 2413-2433, 10.5194/acp-7-2413-2007, 2007.

**Question**: Second, their applications, while interesting, don't well highlight the benefit of the code updates they have implemented. While they do show impacts of updating emissions on the inversion, there is no evaluation of how switching to MERRA2 impacts the inversion, compared to the previously supported GEOS-FP meteorology, despite the fact that they choose an application year (2015) for which both sets of meteorology are available. Also, they've chose to demonstrate the benefits of HEMCO in the adjoint through an off-line CO simulation, which is a linear model for which the inversion results shouldn't depend as strongly on the prior emissions as would e.g. a NOx inversion. There could be some differences based on how the emissions scaling factors are constructed, as examined in recent papers such as Yu et al. (https://doi.org/10.5194/gmd-14-7775-2021), but the authors do not broach this level of detail.

**Answer**: The advantage of the usage of MERRA2 meteorological data is reflected in long-term analysis with consistent meteorological data, which is important for better applications of the adjoint of GEOS-Chem model in the future. As discussed in the revised version of this manuscript: "The adjoint of GEOS-Chem model does not support MERRA-2, and thus, long-term analysis must combine different meteorological reanalysis data, such as GEOS-4 (1985-2007), GEOS-5 (2004-2012) and GEOS-FP (2012-present). For instance, Jiang et al. [2017] constrained global carbon monoxide (CO) emissions in 2001-2015, while the derived trends in CO emissions in *Jiang et al.* [2017] could be affected by the discontinuity among various versions of the meteorological data (i.e., GEOS-4 in 2001-2003, GEOS-5 in 2004-2012 and GEOS-FP in 2013-2015) and the lack of consistency in the model physics of GEOS-5" (*Lines 105-112 in the tracked change file*).

We agree with the reviewer that the advantage of the updated emission inventories should be more significant for full chemistry assimilations. However, it does not mean updated emission inventories are not helpful for tagged-CO assimilations. As discussed in the revised version of this manuscript: "The lack of support to the updated emission inventories can affect the applications of the adjoint of GEOS-Chem model. First, adjoint-based sensitivity analyses are obtained by the backward simulations of atmospheric compositions (i.e., adjoint tracers) and the combination of adjoint tracers with emissions. Out-of-date emission inventories can thus result in inaccurate estimation of the adjoint sensitivities. Second, while inverse analyses are constrained by atmospheric observations, the updated emission inventories are still critical because they are helpful for better convergence of 4D-var assimilations by setting a more reasonable a priori penalty in the cost function. For instance, the a priori biomass burning CO emissions (GFED3, van der Werf et al. (2010)) in Jiang et al. (2017) lack interannual variabilities later than 2011. In order to obtain reasonable convergence of

biomass burning emissions, the a priori biomass burning emissions in September-November 2006 were applied to September-November 2015 over Indonesia in Jiang et al. (2017)" (*Lines 127-148 in the tracked change file*).

Hope the revised manuscript clarifies the advantages of the developed capabilities better. Thank reviewer for pointing out this issue!

Reference:

Jiang, Z., Worden, J. R., Worden, H., Deeter, M., Jones, D. B. A., Arellano, A. F., and Henze, D. K.: A 15-year record of CO emissions constrained by MOPITT CO observations, Atmos Chem Phys, 17, 4565-4583, 10.5194/acp-17-4565-2017, 2017.

van der Werf, G. R., Randerson, J. T., Giglio, L., Collatz, G. J., Mu, M., Kasibhatla, P. S., Morton, D. C., DeFries, R. S., Jin, Y., and van Leeuwen, T. T.: Global fire emissions and the contribution of deforestation, savanna, forest, agricultural, and peat fires (1997–2009), Atmos Chem Phys, 10, 11707-11735, 10.5194/acp-10-11707-2010, 2010.

**Question**: Lastly, the overall gist of this paper in presenting a "new model" runs counter to the community practice for GEOS-Chem and the GEOS-Chem adjoint. The practice of this community is that people use and develop the code, publish their scientific or technical papers on the update, and then these code developments are submitted back to the GEOS-Chem community. Were every code development to constitute a new model, the code base would splinter off into a myriad of different models, hosted by different groups and institutions. GEOS-Chem and its adjoint would not exist without the broader efforts of the community in this regard, including the substantial contributions of co-authors on the present manuscript over the years to the standard GEOS-Chem adjoint code base.

**Answer**: As the reviewer indicated, the development of the adjoint of GEOS-Chem model relies on the contributions from the community. We are very thankful for the help and support from the reviewer in the past years. As shown in the revised manuscript, "The capabilities developed in this work are important for better applications of the adjoint of GEOS-Chem model in the future. These capabilities will be submitted to the standard GEOS-Chem adjoint code base for better development of the community of the adjoint of GEOS-Chem model" (*Lines 29-68 in the tracked change file*).

The title of this manuscript has been changed to: "The capabilities of the adjoint of GEOS-Chem model to support HEMCO emission inventories and MERRA-2 meteorological data". We also made modifications in the text to ensure that the development of these new capabilities is under a unified code base of the adjoint of GEOS-Chem model, for example, "Here we designed a new framework to facilitate emission inventory updates in the adjoint of GEOS-Chem model" (*Lines 14-15 in the tracked change file*) instead of "Here we provide an updated version (GC-Adjoint-HEMCO) of the adjoint of GEOS-Chem model" in the original version. We are sorry for the confusion in the original version of this manuscript!

**Question**: Overall, it is recommended that the authors significantly increase the rigor of their model developments (i.e., test the accuracy of the adjoint sensitivities with respect to emissions in HEMCO, and to transport / convection with MERRA2), and include this as part of the methods or supplemental in a scientifically driven study of CO emissions, using

their new modeling capabilities to address questions of how biases in prior emissions or transport could impact their findings. GMD may not be the best journal for such a paper.

**Answer**: Thank you for the comments! As the answer to the above question, the accuracy of adjoint sensitivities has been tested in the revised manuscript.

Reviewer #1

The manuscript presents an updated version of the adjoint model of the GEOS-Chem chemical transport model. The update supports a new version of the GEOS assimilated meteorology (MERRA-2) and also supports the emission module (HEMCO). The new version of the GEOS-Chem adjoint is then applied to assimilate pseudo-observations and MOPITT observations to constrain anthropogenic CO emissions.

I agree that model developments are crucial and important updates shall be documented in journal papers. However, I feel this manuscript is organized in a simple way: it is not convincing that the new updates are sufficiently important and a GMD paper is needed to document them. Further, it does not demonstrate that the application of the new adjoint can generate new knowledge. I think that the current manuscript needs a major revision to address the two concerns.

**Answer**: Thank you for the comments! The manuscript has been revised based on the comments.

Specific comments:

**Question**: 1) Page 5, Section 2.2. Here it is not clear how the structure of the adjoint code has been updated. Only the way the model reads the emissions? How is that different from the previous adjoint (GC-Adjoint-STD)? As HEMCO has been implemented in the GEOS-Chem forward model, what are the extra efforts to implement it in the adjoint?

**Answer**: As clarified in the revised version: "HEMCO was included in the GEOS-Chem forward simulations in v10-01. HEMCO is responsible for inputs of meteorological and emission data with default support for emission inventories such as CEDS, MIX and NEI2011" (*Lines 114-117 in the tracked change file*). There are noticeable differences between HEMCO and the adjoint of GEOS-Chem model, as the latter is based on GEOS-Chem v8. "First, meteorological and emission data are read with individual modules in the adjoint of GEOS-Chem model. Second, the inputs of emission inventories are undertaken by different modules that were developed individually with significant discrepancies in the source code. In addition, the file format (e.g., binary punch in the adjoint of GEOS-Chem that is the format of older GEOS-Chem versions in contrast to netCDF in HEMCO), emission variables and the usage methods of emission variables (e.g., emission hierarchy, scaling factors and time slice) are inconsistent. These differences have posed a barrier to the application of new emission inventories in the adjoint of GEOS-Chem model" (*Lines 119-126 in the tracked change file*).

"The lack of support to the updated emission inventories can affect the applications of the adjoint of GEOS-Chem model. First, adjoint-based sensitivity analyses are obtained by the backward simulations of atmospheric compositions (i.e., adjoint tracers) and the combination of adjoint tracers with emissions. Out-of-date emission inventories can thus result in inaccurate estimation of the adjoint sensitivities. Second, while inverse analyses are constrained by atmospheric observations, the updated emission inventories are still critical because they are helpful for better convergence of 4D-var assimilations by setting a more reasonable a priori penalty in the cost function" (*Lines 127-144 in the tracked change file*).

"Ideally, people should consider porting the complete HEMCO to the adjoint of GEOS-Chem model to match the new features in GEOS-Chem forward simulations. However, a complete port of HEMCO implies replacing the input framework of the adjoint of GEOS-Chem model, as well as restructuring of HEMCO and the adjoint of GEOS-Chem model to address the compatibility issues, which is very challenging and may not be necessary because the meteorological modules still work well in the adjoint of GEOS-Chem model" (*Lines 149-154 in the tracked change file*).

"Consequently, a major objective of this work is to design a new framework to facilitate emission inventory updates in the adjoint of GEOS-Chem model" (*Lines 154-156 in the tracked change file*). This new framework is not HEMCO, and is different from the original emission inventory modules in the adjoint of GEOS-Chem. "The major advantage of this new framework is good readability and extensibility, which allows us to support HEMCO emission inventories conveniently and to easily add more emissions inventories following future updates in GEOS-Chem forward simulations" (*Lines 15-18 in the tracked change file*).

As indicated by the reviewer, the development in this work is not "Only the way the model reads the emissions". As a 4D-var assimilation system, it is important to ensure consistent emissions in both forward and backward simulations. We have made corresponding modifications to both forward and backward modules, and the reliability of the backward simulations was validated by comparing adjoint gradients of global CO concentrations to CO emissions with finite difference gradients. The capabilities developed in this work are thus reliable and important for better applications of the adjoint of GEOS-Chem model in the future.

**Question**: 2) Section 2.2. Here CO emissions of GC-v12 and GC-Adjoint-HEMCO are compared. Again it is not clear that comparisons of the emissions are sufficiently important to be viewed as a major development. We would expect differences when using different emission inventories, and similar (if not the same) values when using the same emission inventories (as in GC-v12 and GC-Adjoint-HEMCO).

**Answer**: As clarified in the revised version: "In addition to baseline emission data, there are critical factors that affect the usage of emission data in the models. Reading the emission data correctly thus does not necessarily mean using emission data correctly. For example, emission hierarchy is used to prioritize emission fields within the same emission category. Emissions of higher hierarchy overwrite lower hierarchy data. Regional emission inventories usually have a higher hierarchy within their mask boundaries. Scaling factors are used to adjust the baseline emissions with annual, season, month, week, and 24-hour temporal scales. Time slice selection is used to define the usage methods of the emission data outside the original temporal range; for instance, data can be interpreted as climatology and recycled once the end of the last time slice is reached or be only considered as long as the simulation time is within the time range. Furthermore, there are experience parameters applied in files such as emfossil.f and tagged_co.f, which may not be compatible with HEMCO emission inventories. Consequently, we must validate the integrated emissions carefully to ensure that the abovementioned factors have been correctly applied and to ensure that the calculated

emissions are reasonable for individual inventories and the combination of all inventories" (*Lines 278-292 in the tracked change file*).

Furthermore, it should be noted that the comparison of the emissions in Section 2.2 is only the first step of our model evaluations. As clarified in the revised version: "The performances of the developed capabilities were evaluated with the following steps: 1) diagnostic outputs of carbon monoxide (CO) sources and sinks to ensure the correct reading and use of emission inventories; 2) forward simulations to compare the modeled surface and column CO concentrations among various model versions; 3) backward simulations to compare adjoint gradients of global CO concentrations to CO emissions with finite difference gradients; and 4) observing system simulation experiments (OSSE) to evaluate the model performance in 4D variational (4D-var) assimilations" (*Lines 20-27 in the tracked change file*).

The development of the new capabilities in this work is challenging due to the building of an integrated system involving the development of new modules, modifications of existing modules, and usage of various emission data with complex control parameters. Consequently, we have spent great efforts on model evaluation because these evaluations are important to ensure the reliability of the developed capabilities, which is the prerequisite to submitting our update to the standard GEOS-Chem adjoint code base for wider usage by the community.

The discussion in Section 2.2 has been revised. Thank the reviewer for pointing out this issue!

**Question**: 3) Page 10, Section 2.4. Here the authors stated that supporting MERRA-2 is more direct as it can follow the GEOS-FP fields. So how important is the update and any demonstration of that?

**Answer**: The importance of MERRA-2 meteorological data is reflected in long-term analysis with consistent meteorological data. As the reviewer suggested, the manuscript has been revised to clarify this point: "The adjoint of GEOS-Chem model does not support MERRA-2, and thus, long-term analysis must combine different meteorological reanalysis data, such as GEOS-4 (1985-2007), GEOS-5 (2004-2012) and GEOS-FP (2012-present). For instance, Jiang et al. [2017] constrained global carbon monoxide (CO) emissions in 2001-2015, while the derived trends in CO emissions in *Jiang et al.* [2017] could be affected by the discontinuity among various versions of the meteorological data (i.e., GEOS-4 in 2001-2003, GEOS-5 in 2004-2012 and GEOS-FP in 2013-2015) and the lack of consistency in the model physics of GEOS-5" (*Lines 105-112 in the tracked change file*).

This sentence has been revised: "The code porting to support MERRA-2 follows the current framework of the adjoint of GEOS-Chem model … " (*Lines 395-397 in the tracked change file*).

Reference:

Jiang, Z., Worden, J. R., Worden, H., Deeter, M., Jones, D. B. A., Arellano, A. F., and Henze, D. K.: A 15-year record of CO emissions constrained by MOPITT CO observations, Atmos Chem Phys, 17, 4565-4583, 10.5194/acp-17-4565-2017, 2017.

**Question**: 4) Section 3. The section applied the new version of GEOS-Chem adjoint to constrain CO anthropogenic emissions following previous works of the authors. This appears only to show that the adjoint model is running, and does not provide any new scientific findings. Why shall we need to use the new adjoint? Shall we get the same results if we still use the old version (GC-Adjoint-STD) with the prior emissions updated?

**Answer**: As the reviewer indicated, the assimilation experiment in Section 3.3 is designed to show the usability of the developed capabilities, because it is difficult to demonstrate the advantage of GC-Adjoint-HEMCO by performing an assimilation experiment for a single year. In our ongoing project, we are planning to reproduce Jiang et al. [2017] by constraining global CO emissions in 2001-2022 with different observations and OH fields, which is expected to better demonstrate the advantage of the developed capabilities.

A major objective of this work is to design a new framework to facilitate emission inventory updates in the adjoint of GEOS-Chem model. This new framework is not HEMCO, and is different from the original emission inventory modules in the adjoint of GEOS-Chem. The capabilities developed in this work are actually similar to the reviewer's suggestion, i.e., "use the old version (GC-Adjoint-STD) with the prior emissions updated", because the designed new framework provides a convenient pathway to support updated emission inventories. Furthermore, it should be noted that we also developed new modules to support MERRA-2 meteorological data. This allows us to perform long-term analysis with consistent meteorological data in 1979-present, which is not supported by GC-Adjoint-STD.

Reference:

Jiang, Z., Worden, J. R., Worden, H., Deeter, M., Jones, D. B. A., Arellano, A. F., and Henze, D. K.: A 15-year record of CO emissions constrained by MOPITT CO observations, Atmos Chem Phys, 17, 4565-4583, 10.5194/acp-17-4565-2017, 2017.

**Question**: 5) Units are missing for Figures 4 and 5.

**Answer**: The units have been added.

Reviewer #2

The authors developed an updated version (GC-Adjoint-HEMCO) of the adjoint of the GEOS-Chem model to support MERRA-2 meteorological data and HEMCO emission inventories. Their analysis demonstrates good consistency in the forward simulations between GC-Adjoint-HEMCO and standard GEOS-Chem. The reliability of GC-Adjoint-HEMCO in 4D-Var assimilation is further evaluated through observing system simulation experiments (OSSEs). The authors should have spent great efforts on the system development and presented comprehensive results.

This paper is well written. The GC-Adjoint-HEMCO is an important contribution to the community of the adjoint of the GEOS-Chem model. I recommend the paper for publication after consideration of the points below.

**Answer**: Thank you for the comments! As the reviewer indicated, we have spent great efforts on the development of these capabilities. The developed capabilities will be submitted to the standard GEOS-Chem adjoint code base as a part of our contributions to the development of the community of the adjoint of GEOS-Chem model.

Comments:

**Question**: Lines 63-66: It is suggested to provide more discussion to clarify the advantages of the newer emission inventories, as it is the major motivation of the development of GC-Adjoint-HEMCO.

**Answer**: Thank you for this suggestion! As discussed in the revised manuscript: "The lack of support to the updated emission inventories can affect the applications of the adjoint of GEOS-Chem model. First, adjoint-based sensitivity analyses are obtained by the backward simulations of atmospheric compositions (i.e., adjoint tracers) and the combination of adjoint tracers with emissions. Out-of-date emission inventories can thus result in inaccurate estimation of the adjoint sensitivities. Second, while inverse analyses are constrained by atmospheric observations, the updated emission inventories are still critical because they are helpful for better convergence of 4D-var assimilations by setting a more reasonable a priori penalty in the cost function. For instance, the a priori biomass burning CO emissions (GFED3, van der Werf et al. (2010)) in Jiang et al. (2017) lack interannual variabilities later than 2011. In order to obtain reasonable convergence of biomass burning emissions, the a priori biomass burning emissions in September-November 2006 were applied to September-November 2015 over Indonesia in Jiang et al. (2017)" (*Lines 127-148 in the tracked change file*).

Furthermore, it should be noted that the major advantage of our new framework is good readability and extensibility, which not only allows us to support HEMCO emission inventories conveniently, but also allows us to add more emission inventories following future updates in GEOS-Chem forward simulations easily. It is thus important for better applications of the adjoint of GEOS-Chem model in the future.

References:

Jiang, Z., Worden, J. R., Worden, H., Deeter, M., Jones, D. B. A., Arellano, A. F., and Henze, D. K.: A 15-year record of CO emissions constrained by MOPITT CO observations, Atmos Chem Phys, 17, 4565-4583, 10.5194/acp-17-4565-2017, 2017.

van der Werf, G. R., Randerson, J. T., Giglio, L., Collatz, G. J., Mu, M., Kasibhatla, P. S., Morton, D. C., DeFries, R. S., Jin, Y., and van Leeuwen, T. T.: Global fire emissions and the contribution of deforestation, savanna, forest, agricultural, and peat fires (1997–2009), Atmos Chem Phys, 10, 11707-11735, 10.5194/acp-10-11707-2010, 2010.

**Question**: Lines 67-81: It is unclear whether GC-Adjoint-HEMCO can perform assimilations with the full chemistry mode.

**Answer**: As discussed in the revised manuscript: "The capabilities developed in this work are thus based on the tagged-CO mode, as it can effectively accelerate the model development process. More efforts are needed in the future to extend these capabilities to support emissions inventories associated with the full chemistry simulations" (*Lines 178-181 in the tracked change file*). We are sorry for this confusion!

**Question**: Lines 115-118:Please clarify the criteria for assimilation convergence. Why were 40 iterations performed here?

**Answer**: The discussion has been revised: "Following Jiang et al. (2017), we performed 40 iterations (forward + backward simulations) for each month, which usually produced 6-8 accepted iterations (i.e., successful line searches in the large-scale bound constrained optimization (L-BFGS-B, Zhu et al. (1997)) to reduce the cost functions and adjoint gradients. The a posteriori CO emission estimates were calculated based on the last accepted iteration, which usually corresponded to the iteration with the lowest cost function" (*Lines 242-247 in the tracked change file*).

References:

Jiang, Z., Worden, J. R., Worden, H., Deeter, M., Jones, D. B. A., Arellano, A. F., and Henze, D. K.: A 15-year record of CO emissions constrained by MOPITT CO observations, Atmos Chem Phys, 17, 4565-4583, 10.5194/acp-17-4565-2017, 2017.

Zhu, C., Byrd, R. H., Lu, P., and Nocedal, J.: Algorithm 778: L-BFGS-B: Fortran Subroutines for Large-Scale Bound Constrained Optimization, ACM Transactions on Mathematical Software, 23, 550-560, 10.1145/279232.279236, 1997.

**Question**: Line 262-271:The numbers of relative differences are listed in this paragraph. More discussions are suggested to clarify the importance of better consistency between GC-Adjoint-HEMCO and GC-v12.

**Answer**: As the reviewer suggested, the discussion has been revised: "The reasonable emissions in the diagnostic outputs in Section 2 do not necessarily mean the correct integration of emissions in the simulations" (*Lines 423-424 in the tracked change file*). Consequently, we further evaluate the performance of GC-Adjoint-HEMCO in forward simulations. "The agreement between GC-v12 and GC-Adjoint-HEMCO confirms the reliability of GC-Adjoint-HEMCO in forward simulations" (*Lines 438-439 in the tracked change file*).

**Question**: Lines 324-325:It would be better to provide an explanation as why a negative deviation in the initial conditions would lead to an overestimation in the emissions.

**Answer**: Thank you for this suggestion! The discussion has been revised: "the biases in monthly initial CO conditions are caused by model biases in CO concentrations accumulated in previous months. Considering that the lifetime of CO is approximately 2-3 months, the negative biases in the initial conditions can result in negative biases in the modeled CO concentration in the following month. A lack of consideration of these biases, as shown in Fig. 8A, can thus result in overestimations in the derived monthly CO emission estimates because the assimilation system will tend to adjust emissions to reduce the initial condition-induced biases" (*Lines 532-539 in the tracked change file*).

Minors:

**Question**: Line 157-185, it might be worth adding some information (area, seasonal/daily/hourly factors, ) of the different inventorories in Table 1. So the descriptions in the text could be more brief.

**Answer**: Changed.

**Question**: Line 160, mid-week -> weekday?

**Answer**: Changed.

Reviewer #3

This manuscript is presented as a description of a new model / modeling capabilities, in particular running the GEOS-Chem adjoint model using MERRA-2 meteorology, and with emissions processed by HEMCO. While these are valuable developments, and the application shown here with regards to CO emissions estimation is interesting on its own, overall the manuscript is unfortunately problematic for the following reasons.

**Answer**: Thank you for the comments! The manuscript has been revised based on the comments.

**Question**: First, the codes updates included here are alone not significant enough to warrant publication as a stand-alone modeling paper. This is not a new model, just an update to one that is widely used, and not a major update.

**Answer**: As the reviewer indicated, the update here is not a stand-alone model. A major objective of this work is to design a new framework to facilitate emission inventory updates in the widely used adjoint of GEOS-Chem model. It should be noted that this new framework allows direct usage of native HEMCO emission inventories, which is different from the usage of emission data processed by HEMCO, i.e., while experienced users can run HEMCO to output emission data and then develop modules by themselves to read HEMCO-output data into the adjoint of GEOS-Chem, this process is inconvenient and unfriendly, particularly, the HEMCO has to be rerun if there is any change in the inventory settings.

In contrast, the capability to direct usage of native HEMCO emission inventories in this update is much more convenient and friendly to users, which is important for better development of the community of the adjoint of GEOS-Chem model. As shown in the following Figure, the updated emission inventories (such as CEDS, MIX and NEI2011) can be easily selected with simple menu options in the updated input.geos file to keep the same user experience as the standard version of adjoint of GEOS-Chem model. The developed capabilities were clarified in the revised manuscript: "The major advantage of this new framework is good readability and extensibility, which allows us to support HEMCO emission inventories conveniently and to easily add more emissions inventories following future updates in GEOS-Chem forward simulations. Furthermore, we developed new modules to support MERRA-2 meteorological data, which allows us to perform long-term analysis with consistent meteorological data in 1979-present" (*Lines 15-20 in the tracked change file*).

Furthermore, as indicated by the reviewer in the next question, model evaluation is often more time-consuming than code development itself, particularly, the development of the new capabilities in this work is challenging due to the building of an integrated system involving the development of new modules, modifications of existing modules

| Deleted: This |
| Deleted: is not |
| Deleted: , and |
| Deleted: original emission inventory |
| Deleted: in the |
| Deleted: . |

| Deleted: . The capabilities developed in this work are thus important for better applications of the adjoint of GEOS-Chem model in the future. |

and usage of various emission data with complex control parameters. As clarified in the revised version, we have spent great efforts on model evaluation: "The performances of the developed capabilities were evaluated with the following steps: 1) diagnostic outputs of carbon monoxide (CO) sources and sinks to ensure the correct reading and use of emission inventories; 2) forward simulations to compare the modeled surface and column CO concentrations among various model versions; 3) backward simulations to compare adjoint gradients of global CO concentrations to CO emissions with finite difference gradients; and 4) observing system simulation experiments (OSSE) to evaluate the model performance in 4D variational (4D-var) assimilations" (*Lines 20-27 in the tracked change file*). The capabilities developed in this work are thus reliable and important for better applications of the adjoint of GEOS-Chem model in the future.

```
%%% EMISSIONS MENU %%%  :
Turn on emissions?      : T
Emiss timestep (min)    : 20
Include anthro emiss?   : T
 => Scale to (1985-2005): -1
 => Use EMEP emissions? : F
 => Use BRAVO emissions?: F
 => Use EDGAR emissions?: F
 => Use CEDS emissions? : T
 => Use MIX  emiss?     : T
 => Use APEI emissions? : T
 => Use NEI2005 emiss?  : F
 => Use NEI2011 emiss?  : T
 => Use RETRO emiss?    : F
 => Use RCP   emiss?    : F
        RCP scenario ?  : RCP60
        RCP year ?      : 2010
 => Use AFRICA emiss?   : T
```

**Question**: Second, the authors only validate one aspect of their model updats — the forward model performance — but do not evaluate nor validate the adjoint code. This is a good sanity check, but not sufficient for a publication, nor for demonstration that their updates are correct and complete. As adjoint code developers well know, it is the verification of the adjoint sensitivities following any code update which is often more time consuming than the forward model update. However, the authors have not demonstrated that adjoint sensitivities are correctly propagated through HEMCO, or when running MERRA-2 meteorology. This needs to be demonstrated via numerical evaluation of the accuracy of their adjoint sensitivities for horizontal advection, vertical convection, and emissions scaling factors. This is a standard test for any update to the GEOS-Chem adjoint, and their code is not ready for submission to the community code repository until this has been demonstrated. If the updates are so trivial that the authors feel numerical tests are not required, this would further emphasize that the work, while commendable, is not substantial enough to warrant a stand-alone publication.

**Answer**: New Fig. 6 was added in the revised version to show the comparison of adjoint and finite difference gradients of global CO concentrations to CO emission, by turning on the convection, planetary boundary layer mixing and advection processes

individually, and the effects of combined model processes with various assimilation windows (turning off advection as suggested by Henze et al. (2007)). We find good agreement between the adjoint and finite difference gradients. This confirms the consistency between forward and backward simulations in the developed capabilities.

Reference:

Henze, D. K., Hakami, A., and Seinfeld, J. H.: Development of the adjoint of GEOS-Chem, Atmos Chem Phys, 7, 2413-2433, 10.5194/acp-7-2413-2007, 2007.

**Question**: Second, their applications, while interesting, don't well highlight the benefit of the code updates they have implemented. While they do show impacts of updating emissions on the inversion, there is no evaluation of how switching to MERRA2 impacts the inversion, compared to the previously supported GEOS-FP meteorology, despite the fact that they choose an application year (2015) for which both sets of meteorology are available. Also, they've chose to demonstrate the benefits of HEMCO in the adjoint through an off-line CO simulation, which is a linear model for which the inversion results shouldn't depend as strongly on the prior emissions as would e.g. a NOx inversion. There could be some differences based on how the emissions scaling factors are constructed, as examined in recent papers such as Yu et al. (https://doi.org/10.5194/gmd-14-7775-2021), but the authors do not broach this level of detail.

**Answer**: The advantage of the usage of MERRA2 meteorological data is reflected in long-term analysis with consistent meteorological data, which is important for better applications of the adjoint of GEOS-Chem model in the future. As discussed in the revised version of this manuscript: "The adjoint of GEOS-Chem model does not support MERRA-2, and thus, long-term analysis must combine different meteorological reanalysis data, such as GEOS-4 (1985-2007), GEOS-5 (2004-2012) and GEOS-FP (2012-present). For instance, Jiang et al. [2017] constrained global carbon monoxide (CO) emissions in 2001-2015, while the derived trends in CO emissions in *Jiang et al.* [2017] could be affected by the discontinuity among various versions of the meteorological data (i.e., GEOS-4 in 2001-2003, GEOS-5 in 2004-2012 and GEOS-FP in 2013-2015) and the lack of consistency in the model physics of GEOS-5" (*Lines 105-112 in the tracked change file*).

We agree with the reviewer that the advantage of the updated emission inventories should be more significant for full chemistry assimilations. However, it does not mean updated emission inventories are not helpful for tagged-CO assimilations. As discussed in the revised version of this manuscript: "The lack of support to the updated emission inventories can affect the applications of the adjoint of GEOS-Chem model. First, adjoint-based sensitivity analyses are obtained by the backward simulations of atmospheric compositions (i.e., adjoint tracers) and the combination of adjoint tracers with emissions. Out-of-date emission inventories can thus result in inaccurate estimation of the adjoint sensitivities. Second, while inverse analyses are constrained by atmospheric observations, the updated emission inventories are still critical because they are helpful for better convergence of 4D-var assimilations by setting a more reasonable a priori penalty in the cost function. For instance, the a priori biomass burning CO emissions (GFED3, van der Werf et al. (2010)) in Jiang et al. (2017) lack interannual variabilities later than 2011. In order to obtain reasonable convergence of

biomass burning emissions, the a priori biomass burning emissions in September-November 2006 were applied to September-November 2015 over Indonesia in Jiang et al. (2017)" (*Lines 127-148 in the tracked change file*).

Hope the revised manuscript clarifies the advantages of the developed capabilities better. Thank reviewer for pointing out this issue!

Reference:

Jiang, Z., Worden, J. R., Worden, H., Deeter, M., Jones, D. B. A., Arellano, A. F., and Henze, D. K.: A 15-year record of CO emissions constrained by MOPITT CO observations, Atmos Chem Phys, 17, 4565-4583, 10.5194/acp-17-4565-2017, 2017.

van der Werf, G. R., Randerson, J. T., Giglio, L., Collatz, G. J., Mu, M., Kasibhatla, P. S., Morton, D. C., DeFries, R. S., Jin, Y., and van Leeuwen, T. T.: Global fire emissions and the contribution of deforestation, savanna, forest, agricultural, and peat fires (1997–2009), Atmos Chem Phys, 10, 11707-11735, 10.5194/acp-10-11707-2010, 2010.

**Question**: Lastly, the overall gist of this paper in presenting a "new model" runs counter to the community practice for GEOS-Chem and the GEOS-Chem adjoint. The practice of this community is that people use and develop the code, publish their scientific or technical papers on the update, and then these code developments are submitted back to the GEOS-Chem community. Were every code development to constitute a new model, the code base would splinter off into a myriad of different models, hosted by different groups and institutions. GEOS-Chem and its adjoint would not exist without the broader efforts of the community in this regard, including the substantial contributions of co-authors on the present manuscript over the years to the standard GEOS-Chem adjoint code base.

**Answer**: As the reviewer indicated, the development of the adjoint of GEOS-Chem model relies on the contributions from the community. We are very thankful for the help and support from the reviewer in the past years. As shown in the revised manuscript, "The capabilities developed in this work are important for better applications of the adjoint of GEOS-Chem model in the future. These capabilities will be submitted to the standard GEOS-Chem adjoint code base for better development of the community of the adjoint of GEOS-Chem model" (*Lines 29-68 in the tracked change file*).

The title of this manuscript has been changed to: "The capabilities of the adjoint of GEOS-Chem model to support HEMCO emission inventories and MERRA-2 meteorological data". We also made modifications in the text to ensure that the development of these new capabilities is under a unified code base of the adjoint of GEOS-Chem model, for example, "Here we designed a new framework to facilitate emission inventory updates in the adjoint of GEOS-Chem model" (*Lines 14-15 in the tracked change file*) instead of "Here we provide an updated version (GC-Adjoint-HEMCO) of the adjoint of GEOS-Chem model" in the original version. We are sorry for the confusion in the original version of this manuscript!

**Question**: Overall, it is recommended that the authors significantly increase the rigor of their model developments (i.e., test the accuracy of the adjoint sensitivities with respect to emissions in HEMCO, and to transport / convection with MERRA2), and include this as part of the methods or supplemental in a scientifically driven study of CO emissions, using

their new modeling capabilities to address questions of how biases in prior emissions or transport could impact their findings. GMD may not be the best journal for such a paper.

**Answer**: Thank you for the comments! As the answer to the above question, the accuracy of adjoint sensitivities has been tested in the revised manuscript.

---

## Author Response (AR3)

We thank the reviewer for the thoughtful and detailed comments. We have revised this manuscript carefully based on the comments. Below we respond to the individual comments:

Reviewer #3, Prof. Daven Henze

**Question**: The authors have taken some care to respond to the reviewer questions, largely to further explain why they feel the incorporation of HEMCO into the GEOS-Chem adjoint model is a substantial enough update to warrant a publication. They have also further evaluated their update in terms of testing the adjoint gradients with respect to emissions that are now processed by HEMCO, and the performance is perfect (see the new Fig. 6), thus I'm convinced the update works as intended.

That being said, I still have reservations regarding this work as a whole being the basis of a publication, given that it was only upon reading their response to reviewer's question that it became clear to me their updates are only for the tagged-CO simulation and not the full-chemistry simulation. The potential impact and audience is thus smaller. Also, the development of support for MERRA2 meteorology has already been done for non-full-chemistry simulations that have been in place within the standard GEOS-Chem adjoint for up to seven years — that of the CH4 simulation and N2O simulations. They have though made clear that they plan to submit their code updates to the standard adjoint code basis, and perhaps in this process the work can be expanded to support simulation types beyond the tagged-CO.

**Answer**: Thank the reviewer for the comments! The code update will be submitted to the standard adjoint code basis once the extension to support more simulation types is finished.

Some specific comments on their revisions:

**Question**: For equations (3) and (4) of the revised manuscript, one shows the sensitivity with respect to a variable 'x' and the other to a variable 'sigma' yet the point here is to provide two different equations (methods) for computing the same sensitivities, which can then be compared as a validation. I thus suggest the nomenclature be adjusted such that the adjoint and finite difference sensitivities are with respect to the same variable (x, or sigma).

**Answer**: The equations have been revised by using the same variable x.

**Question**: In the caption for Figure 6, they refer to sensitivities of "global CO concentrations" but I think rather they mean sensitivities of CO in individual grid boxes or columns. In this way one can perform the ensemble of necessary finite difference and adjoint calculations in parallel.

**Answer**: The finite difference experiments shown in Fig. 6 are performed with the "LFD_GLOB" option (global perturbations) and "LFD=1" (model level 1). Consistent results were obtained by using larger LFD numbers (i.e., higher model levels within the PBL and free troposphere) as shown in new Fig. S10 and S11. The caption of Fig. 6 was updated to clarify that they are the "sensitivities of global CO concentrations (LFD_GLOB and model level 1)".

**Question**: On line 196 of the revised manuscript, they mention ``experience parameters''
— it's not clear to me what this means. Is this a typo, or could they explain more?

**Answer**: There are some discrepancies in the treatment of emission data between GC-Adjoint-STD and GC-v12. For example, anthropogenic CO emissions are enhanced by 19% in tagged-CO simulation in GC-Adjoint-STD to account for CO production from anthropogenic VOC. However, the application of this factor is removed in tagged-CO simulation in GC-v12. We have checked the usage of such parameters to ensure the consistency between GC-Adjoint-HEMCO and GC-v12.

As indicated by the reviewer, the description of "experience parameters" in the original version is unclear and we find that it may not be useful to show too many details about the source code. This sentence has thus been deleted in the revision.